# Mitigating Manipulation in Peer Review via Randomized Reviewer Assignments

**Steven Jecmen**
Carnegie Mellon University
sjecmen@cs.cmu.edu

**Hanrui Zhang**
Duke University
hrzhang@cs.duke.edu

**Ryan Liu**
Carnegie Mellon University
ryanliu@andrew.cmu.edu

**Nihar B. Shah**
Carnegie Mellon University
nihars@cs.cmu.edu

**Vincent Conitzer**
Duke University
conitzer@cs.duke.edu

**Fei Fang**
Carnegie Mellon University
feif@cs.cmu.edu

## Abstract

We consider three important challenges in conference peer review: (i) reviewers maliciously attempting to get assigned to certain papers to provide positive reviews, possibly as part of quid-pro-quo arrangements with the authors; (ii) "torpedo reviewing," where reviewers deliberately attempt to get assigned to certain papers that they dislike in order to reject them; (iii) reviewer de-anonymization on release of the similarities and the reviewer-assignment code. On the conceptual front, we identify connections between these three problems and present a framework that brings all these challenges under a common umbrella. We then present a (randomized) algorithm for reviewer assignment that can optimally solve the reviewer-assignment problem under any given constraints on the probability of assignment for any reviewer-paper pair. We further consider the problem of restricting the joint probability that certain suspect pairs of reviewers are assigned to certain papers, and show that this problem is NP-hard for arbitrary constraints on these joint probabilities but efficiently solvable for a practical special case. Finally, we experimentally evaluate our algorithms on datasets from past conferences, where we observe that they can limit the chance that any malicious reviewer gets assigned to their desired paper to $50\%$ while producing assignments with over $90\%$ of the total optimal similarity.

## 1 Introduction

Peer review, the evaluation of work by others working in the same field as the producer of the work or with similar competencies, is a critical component of scientific research. It is regarded favorably by a significant majority of researchers and is seen as being essential to both improving the quality of published research and validating the legitimacy of research publications [1–3]. Due to the wide adoption of peer review in the publication process in academia, the peer-review process can be very high-stakes for authors, and the integrity of the process can significantly influence the careers of the authors (especially due to the prominence of a "rich get richer" effect in academia [4]).

However, there are several challenges that arise in peer review relating to the integrity of the review process. In this work, we address three such challenges for peer review in academic conferences where a number of papers need to be assigned to reviewers at the same time.

**(1) Untruthful favorable reviews.** In order to achieve a good reviewer assignment, peer review systems must solicit some information about their reviewers' knowledge and interests (e.g., through bidding). This can be manipulated, and in fact this is known to have happened in at least one ACM

conference [5]: "*Another SIG community has had a collusion problem where the investigators found that a group of PC members and authors colluded to bid and push for each other's papers violating the usual conflict-of-interest rules.*" The problem of manipulation is not limited to the bidding system, as practically anything used to determine paper assignments (e.g., self-reported area of expertise, list of papers the reviewer has published) can potentially be manipulated [6, 7]. In some cases, unethical authors may enter into deals with potential reviewers for their paper, where the reviewer agrees to attempt to get assigned to the author's paper and give it a favorable review in exchange for some outside reward (e.g., as part of a quid-pro-quo arrangement for the reviewer's own paper in another publication venue). To preserve the integrity of the reviewing process and maintain community trust, the paper assignment algorithm should guarantee the mitigation of these kinds of arrangements.

**(2) Torpedo reviewing.** In "torpedo reviewing," unethical reviewers attempt to get assigned to papers they dislike with the intent of giving them an overly negative review and blocking the paper from publication. This can have wide-reaching consequences [8]: "*Repeated indefinitely, this gives the power to kill off new lines of research to the 2 or 3 most close-minded members of a community, potentially substantially retarding progress for the community as a whole.*" One special case of torpedo reviewing has been called "rational cheating," referring to reviewers negatively reviewing papers that compete with their own authored work [9, 10]. A paper assignment algorithm should guarantee to authors that their papers are unlikely to have been torpedo-reviewed.

**(3) Reviewer de-anonymization in releasing assignment data.** For transparency and research purposes, conferences may wish to release the paper-reviewer similarities and the paper assignment algorithm used after the conference. However, if the assignment algorithm is deterministic, this would allow for authors to fully determine who reviewed their paper, breaking the anonymity of the reviewing process. Even when reviewer and paper names are removed, identities can still be discovered (as in the case of the Netflix Prize dataset [11]). Consequently, a rigorous guarantee of anonymity is needed in order to release the data.

Although these challenges may seem disparate, we address all of them under a common umbrella framework. Our contributions are as follows. **Conceptually**, we formulate problems concerning the three aforementioned issues, and propose a framework to address them through the use of randomized paper assignments (Section 3). **Theoretically**, we design computationally efficient, randomized assignment algorithms that optimally assign reviewers to papers subject to given restrictions on the probability of assigning any particular reviewer-paper pair (Section 4). We further consider the more complex case of preventing suspicious *pairs* of reviewers from being assigned to the same paper (Section 5). We show that finding the optimal assignment subject to arbitrary constraints on the probabilities of reviewer-reviewer-paper assignments is NP-hard. In the practical special case where the program chairs want to prevent pairs of reviewers within the same subset of some partition of the reviewer set (e.g., by academic institution) from being assigned to the same paper, we present an algorithm that finds the optimal randomized assignment with this guarantee. **Empirically**, we test our algorithms on datasets from past conferences and show their practical effectiveness (Section 6). As a representative example, on data reconstructed from ICLR 2018, our algorithms can limit the chance of any reviewer-paper assignment to $50\%$ while achieving $90.8\%$ of the optimal total similarity. Our algorithms can continue to achieve this similarity while also preventing reviewers with close associations from being assigned to the same paper. We further demonstrate, using the ICLR 2018 dataset, that our algorithm successfully prevents manipulation of the assignment by a simulated malicious reviewer.

All of the code for our algorithms and our empirical results is freely available online.[1]

## 2   Related Literature

Many paper assignment algorithms for conference peer review have been proposed in past work. The widely-used Toronto Paper Matching System (TPMS) [12] computes a similarity score for each reviewer-paper pair based on analysis of the reviewers' past work and bids, and then aims to maximize the total similarity of the resulting assignment. The framework of "compute similarities and maximize total similarity" (and similar variants) encompasses many paper assignment algorithms, where similarities can be computed in various ways from automated and manual analysis and

reviewer bids [13–19]. We treat the bidding process and computation of similarities as given, and focus primarily on adjusting the optimization problem to address the three aforementioned challenges. There are also a number of recent works [20–34] which deal with other aspects of peer review.

Much prior work has studied the issue of preventing or mitigating strategic behavior in peer review. This work usually focuses on the incentives reviewers have to give poor reviews to other papers in the hopes of increasing their own paper's chances of acceptance [35–39]. Unlike the issues we deal with in this paper, these works consider only reviewers' incentives to get *their own paper accepted* and not other possible incentives. We instead consider arbitrary incentives for a reviewer to give an untruthful review, such as a personal dislike for a research area or misincentives brought about by author-reviewer collusion. Instead of aiming to remove reviewer incentives to be untruthful, our work focuses on mitigating the effectiveness of manipulating the reviewer assignment process.

Randomized assignments have been used to address the problem of fair division of indivisible goods such as jobs or courses [40, 41], as well as in the context of Stackelberg security games [42]. The paper [22] uses randomization to address the issue of miscalibration in ratings, such as those given to papers in peer review. To the best of our knowledge, the use of randomized reviewer-paper assignments to address the issues of malicious reviewers or reviewer de-anonymization in peer review has not been studied previously.

Additional coverage of related literature is presented in Appendix A.

## 3 Background and Problem Statements

We first define the standard paper assignment problem, followed by our problem setting. In the standard paper assignment setting, we are given a set $\mathcal{R}$ of $n$ reviewers and a set $\mathcal{P}$ of $d$ papers, along with desired reviewer load $k$ (that is, the maximum number of papers any reviewer should be assigned) and desired paper load $\ell$ (that is, the exact number of reviewers any paper should be assigned to).[2] An assignment of papers to reviewers is a bipartite matching between the sets that obeys the load constraints on all reviewers and papers. In addition, we are given a similarity matrix $S \in \mathbb{R}^{n \times d}$ where $S_{rp}$ denotes how good of a match reviewer $r$ is for paper $p$. These similarities can be derived from the reviewers' bids on papers, prior publications, conflicts of interest, etc.

The standard problem of finding a maximum sum-similarity assignment [12, 13, 15, 17, 19] is then written as an integer linear program. The decision variables $M \in \{0, 1\}^{n \times d}$ specify the assignment, where $M_{rp} = 1$ if and only if reviewer $r$ is assigned to paper $p$. The objective is to maximize $\sum_{r \in \mathcal{R}} \sum_{p \in \mathcal{P}} S_{rp} M_{rp}$ subject to the load constraints $\sum_{p \in \mathcal{P}} M_{rp} \leq k, \forall r \in \mathcal{R}$ and $\sum_{r \in \mathcal{R}} M_{rp} = \ell, \forall p \in \mathcal{P}$. Since the constraint matrix of the linear program (LP) relaxation of this problem is totally unimodular, the solution to the LP relaxation will be integral and so this problem can be solved as an LP. This method of assigning papers has been used by various conferences such as NeurIPS, ICML, ICCV, and SIGKDD (among others) [12, 17], as well as by popular conference management systems EasyChair (`easychair.org`) and HotCRP (`hotcrp.com`).

When the assignment algorithm is deterministic in this way, a malicious reviewer who knows the algorithm may be able to effectively manipulate it in order to get assigned to a desired paper. To address this issue, we aim to provide a guarantee that regardless of the reviewer bids and similarities, this reviewer-paper pair has only a limited probability of being assigned. Additionally, consider the challenge of preserving anonymity in releasing conference data. If the assignment algorithm provides a guarantee that each reviewer-paper pair has only a limited probability of being assigned, then no reviewer's identity can be discovered with certainty. With this motivation, we now consider $M$ as stochastic and aim to find a *randomized assignment*, a probability distribution over deterministic assignments. This naturally leads to the following problem formulation.

**Definition 1** (Pairwise-Constrained Problem)**.** *The input to the problem is a similarity matrix $S$ and a matrix $Q \in [0, 1]^{n \times d}$. The goal is to find a randomized assignment of papers to reviewers (i.e., a distribution of $M$) that maximizes $\mathbb{E}\left[\sum_{r \in \mathcal{R}} \sum_{p \in \mathcal{P}} S_{rp} M_{rp}\right]$ subject to the constraints $\mathbb{P}[M_{rp} = 1] \leq Q_{rp}, \forall r \in \mathcal{R}, p \in \mathcal{P}$.*

Since a randomized assignment is a distribution over deterministic assignments, all assignments $M$ in the support of the randomized assignment must still obey the load constraints $\sum_{p \in \mathcal{P}} M_{rp} \leq k, \forall r \in \mathcal{R}$ and $\sum_{r \in \mathcal{R}} M_{rp} = \ell, \forall p \in \mathcal{P}$. The optimization objective is the expected sum-similarity across all paper-reviewer pairs, the natural analogue of the deterministic sum-similarity objective.

To prevent dishonest reviews of papers, program chairs may want to do more than just control the probability of individual paper-reviewer pairs. For example, suppose that we have three reviewers assigned per paper (a very common arrangement in computer science conferences). We might not be particularly concerned about preventing any single reviewer from being assigned to some paper, since even if that reviewer dishonestly reviews the paper, there are likely two other honest reviewers who can overrule the dishonest one. However, it would be much worse if we have two reviewers dishonestly reviewing the same paper, since they could likely overrule the sole honest reviewer.

A second issue is that there may be dependencies within certain pairs of reviewers that cannot be accurately represented by constraints on individual reviewer-paper pairs. For example, we may have two reviewers $a$ and $b$ who are close collaborators, each of which we are not individually very concerned about assigning to paper $p$. However, we may believe that in the case where reviewer $a$ has entered into a quid-pro-quo deal to dishonestly review paper $p$, reviewer $b$ is likely to also be involved in the same deal. Therefore, one may want to strictly limit the probability that **both** reviewers $a$ and $b$ are assigned to paper $p$, regardless of the limits on the probability that either reviewer individually is assigned to paper $p$.

With this motivation, we define the following generalization of the Pairwise-Constrained Problem.

**Definition 2** (Triplet-Constrained Problem). *The input to the problem is a similarity matrix $S$, a matrix $Q \in [0,1]^{n \times d}$, and a 3-dimensional tensor $T \in [0,1]^{n \times n \times d}$. The goal is to find a randomized assignment of papers to reviewers that maximizes $\mathbb{E}\left[\sum_{r \in \mathcal{R}} \sum_{p \in \mathcal{P}} S_{rp} M_{rp}\right]$ subject to the constraints $\mathbb{P}[M_{rp} = 1] \leq Q_{rp}, \forall r \in \mathcal{R}, p \in \mathcal{P}$ and $\mathbb{P}[M_{ap} = 1 \wedge M_{bp} = 1] \leq T_{abp}, \forall a, b \in \mathcal{R}$ s.t. $a \neq b, p \in \mathcal{P}$.*

The randomized assignments that solve these problems can be used to address all three challenges we identified earlier:

- **Untruthful favorable reviews:** By guaranteeing a limit on the probability that any malicious reviewer or pairs of reviewers can be assigned to the paper they want, we mitigate the effectiveness of any unethical deals between reviewers and authors by capping the probability that such a deal can be upheld. This guarantee holds regardless of how extreme a reviewers' manipulation of the assignment is and without any assumptions on reviewers' exact incentives. The entries of $Q$ or $T$ can be set by the program chairs based on their assessment of the risk of allowing the corresponding assignment.

- **Torpedo reviewing:** By limiting the probability that any reviewer or pair of reviewers can be assigned to a paper they wish to torpedo, we make it much more difficult for a small group of reviewers to shut down a new research direction or to take out competing papers.

- **Reviewer de-anonymization in releasing assignment data:** If all of the entries in $Q$ are set to some reasonable constant value, only the distribution over assignments can be recovered and not the specific assignment that was actually used. This guarantees that for each paper, no reviewer's identity can be identified with high confidence, since every reviewer has only a limited chance to be assigned to that paper.

In Sections 4 and 5, we consider the Pairwise-Constrained Problem and Triplet-Constrained Problem respectively. We also consider several related problems in Appendices B and C.

## 4 Randomized Assignment with Reviewer-Paper Constraints

In this section we present our main algorithm to solve the Pairwise-Constrained Problem (Definition 1), thereby addressing the challenges identified earlier. Before delving into the details of the algorithm, the following theorem states our main result.

**Theorem 1.** *There exists an algorithm which returns an optimal solution to the Pairwise-Constrained Problem in $poly(n, d)$ time.*

We describe the algorithm, thereby proving this result, in the next two subsections. Our algorithm that realizes this result has two parts. In the first part, we find an optimal "fractional assignment matrix," which gives the marginal probabilities of individual reviewer-paper assignments. The second part of the algorithm then samples an assignment, respecting the marginal probabilities specified by this fractional assignment.

## 4.1 Finding the Fractional Assignment

Define a *fractional assignment matrix* as a matrix $F \in [0, 1]^{n \times d}$ that obeys the load constraints $\sum_{p \in \mathcal{P}} F_{rp} \leq k$ for all reviewers $r \in \mathcal{R}$ and $\sum_{r \in \mathcal{R}} F_{rp} = \ell$ for all papers $p \in \mathcal{P}$. Note that any deterministic assignment can be represented by a fractional assignment matrix with all entries in $\{0, 1\}$. Any randomized assignment is associated with a fractional assignment matrix where $F_{rp}$ is the marginal probability that reviewer $r$ is assigned to paper $p$. Furthermore, randomized assignments associated with the same fractional assignment matrix have the same expected sum-similarity. The paper [43] proves an extension of the Birkhoff-von Neumann theorem [44, 45] which shows that all fractional assignment matrices are implementable, i.e., they are associated with at least one randomized assignment. On the other hand, any probability matrix not obeying the load constraints cannot be implemented by a lottery over deterministic assignments, since all deterministic assignments do obey the constraints. Therefore, finding the optimal randomized assignment is equivalent to solving the following LP, which we call $\mathcal{LP}1$:

$$\underset{F \in \mathbb{R}^{n \times d}}{\arg\max} \quad \sum_{p \in \mathcal{P}} \sum_{r \in \mathcal{R}} S_{rp} F_{rp} \tag{1}$$

$$\text{subject to} \quad 0 \leq F_{rp} \leq 1 \qquad\qquad \forall r \in \mathcal{R}, \forall p \in \mathcal{P} \tag{2}$$

$$\sum_{p \in \mathcal{P}} F_{rp} \leq k \qquad\qquad \forall r \in \mathcal{R} \tag{3}$$

$$\sum_{r \in \mathcal{R}} F_{rp} = \ell \qquad\qquad \forall p \in \mathcal{P} \tag{4}$$

$$F_{rp} \leq Q_{rp} \qquad\qquad \forall r \in \mathcal{R}, \forall p \in \mathcal{P}. \tag{5}$$

$\mathcal{LP}1$ has $O(dn)$ variables and constraints. By [46], $\mathcal{LP}1$ can be solved in $O((dn)^{2.055})$ time.

## 4.2 Implementing the Probabilities

$\mathcal{LP}1$ only finds the optimal marginal assignment probabilities $F$ (where $F$ now refers to a solution to $\mathcal{LP}1$). It remains to show whether and how these marginal probabilities can be implemented as a randomization over deterministic paper assignments. The paper [43] provides a method for sampling a deterministic assignment from a fractional assignment matrix, which completes our algorithm once applied to the optimal solution of $\mathcal{LP}1$. Here we briefly sketch a simpler version of the sampling algorithm. In Appendix D, we describe the sampling algorithm in detail along with a supplementary algorithm to compute the full distribution over assignments.

The algorithm begins by constructing a complete bipartite graph with vertices representing reviewers and papers, and connecting source and sink vertices to the reviewer and paper vertices respectively. It also constructs a flow function over the edges, where the initial flow on each reviewer-paper edge is set equal to the marginal assignment probability for the adjacent reviewer and paper. The algorithm then proceeds in an iterative manner, modifying the flow function on each iteration. On each iteration, we first check if there exists a "fractional edge," an edge with non-integral flow. If no such edge exists, we can stop iterating and return the assignment represented by the current integral flows. If there does exist a fractional edge, we then find an arbitrary cycle of fractional edges, ignoring direction. On finding a cycle, we modify the flow function by "pushing" flow along the cycle in either the forwards or backwards direction until the flow on some edge becomes integral. We randomly choose the direction with probabilities proportional to the amount of flow pushed, so that the final sampled assignment satisfies the desired marginal probabilities.

Each iteration of this algorithm takes $O(d + n)$ time to find a cycle in the $O(d + n)$ vertices (if a list of fractional edges adjacent to each vertex is maintained), and it can take $O(dn)$ iterations to terminate since one edge becomes integral every iteration. Therefore, the sampling algorithm

is overall $O(dn(d + n))$. The time complexity of our full algorithm, including both $\mathcal{LP}1$ and the sampling algorithm, is dominated by the complexity of solving the LP. Since standard paper assignment algorithms such as TPMS can be implemented by solving an LP of the same size, our algorithm is comparable in complexity. If a conference currently does solve an LP to find their assignment, whatever LP solver a conference currently uses for their paper assignment algorithm could be used in our algorithm as well.

## 5 Randomized Assignment with Constraints on Pairs of Reviewers

We now turn to the problem of controlling the probabilities that certain pairs of reviewers are assigned to the same paper, defined in Section 3 as the Triplet-Constrained Problem (Definition 2). As described in Section 3, solving the Triplet-Constrained Problem would allow the program chairs of a conference maximum flexibility in how they control the probabilities of the assignments of pairs of reviewers. Unfortunately, as the following theorem shows, this problem cannot be efficiently solved. The proof is stated in Appendix E.

**Theorem 2.** *The Triplet-Constrained Problem is NP-hard, by reduction from 3-Dimensional Matching.*

Since the most general problem of arbitrary constraints on reviewer-reviewer-paper triples is NP-hard, we must restrict ourselves to tractable special cases of interest. One such special case arises when the program chairs of a conference can partition the reviewers in such a way that they wish to prevent any two reviewers within the same subset from being assigned to the same paper. For example, reviewers can be partitioned by their primary academic institution. Since reviewers at the same institution are likely closely associated, program chairs may believe that placing them together as co-reviewers is more risky than would be implied by our concern about either reviewer individually. In this case, there may not even be any concern about the reviewers' motivations; the concern may simply be that the reviewers' opinions would not be sufficiently independent. Other partitions of interest could be the reviewer's geographical area of residence or research sub-field, as each of these defines a "community" of reviewers that may be more closely associated. This special case corresponds to instances of the Triplet-Constrained Problem where $T_{abp} = 0$ if reviewers $a$ and $b$ are in the same subset, and $T_{abp} = 1$ otherwise. We formally define this problem as follows:

**Definition 3** (Partition-Constrained Problem). *The input to the problem is a similarity matrix $S$, a matrix $Q \in [0, 1]^{n \times d}$, and a partition of the reviewer set into subsets $I_1, \ldots, I_m \subseteq \mathcal{R}$. The goal is to find a randomized assignment of papers to reviewers that maximizes $\mathbb{E}\left[\sum_{r \in \mathcal{R}} \sum_{p \in \mathcal{P}} S_{rp} M_{rp}\right]$ subject to the constraints that $\mathbb{P}[M_{rp} = 1] \leq Q_{rp}, \forall r \in \mathcal{R}, p \in \mathcal{P}$, and $\mathbb{P}[M_{ap} = 1 \wedge M_{bp} = 1] = 0, \forall a, b \in I_i, \forall i \in [m]$.*

For this special case of the Triplet-Constrained Problem, we show that the problem is efficiently solvable, as stated in the following theorem.

**Theorem 3.** *There exists an algorithm which returns an optimal solution to the Partition-Constrained Problem in poly(n, d) time.*

We present the algorithm that realizes this result in the following subsections, thus proving the theorem. The algorithm has two parts: it first finds a fractional assignment matrix $F$ meeting certain requirements, and then samples an assignment while respecting the marginal assignment probabilities given by $F$ and additionally never assigning two reviewers from the same subset to the same paper. For ease of exposition, we first present the sampling algorithm, and then present an LP which finds the optimal fractional assignment matrix meeting the necessary requirements.

### 5.1 Partition-Constrained Sampling Algorithm

The sampling algorithm we present in this section takes as input a fractional assignment matrix $F$ and samples an assignment while respecting the marginal assignment probabilities given by $F$. The sampling algorithm is based on the following lemma:

**Lemma 1.** *Consider any fractional assignment matrix $F$ and any partition of $\mathcal{R}$ into subsets $I_1, \ldots, I_m$.*

*(i) There exists a sampling algorithm that implements the marginal assignment probabilities given by F and runs in $O(dn(d+n))$ time such that, for all papers $p \in \mathcal{P}$ and subsets $I \in \{I_1, \ldots, I_m\}$ where $\sum_{r \in I} F_{rp} \leq 1$, the algorithm never samples an assignment assigning two reviewers from subset $I$ to paper $p$.*

*(ii) For any sampling algorithm that implements the marginal assignment probabilities given by F, for all papers $p \in \mathcal{P}$ and subsets $I \in \{I_1, \ldots, I_m\}$ where $\sum_{r \in I} F_{rp} > 1$, the expected number of pairs of reviewers from subset $I$ assigned to paper $p$ is strictly positive.*

The algorithm which realizes Lemma 1 has an additional property, which holds *simultaneously* for all papers and subsets. We state the property in the following corollary and make use of it later:

**Corollary 1.** *For any fractional assignment matrix F, the sampling algorithm that realizes Lemma 1 minimizes the expected number of pairs of reviewers from subset $I$ assigned to paper $p$ simultaneously for all papers $p \in \mathcal{P}$ and subsets $I \in \{I_1, \ldots, I_m\}$ among all sampling algorithms implementing the marginal assignment probabilities given by F.*

We briefly sketch the sampling algorithm that realizes these results here, and describe it in detail along with proofs of the guarantees stated in Lemma 1 and Corollary 1 in Appendix F. This algorithm is a modification of the sampling algorithm from Theorem 1 sketched earlier in Section 4.2. We first provide some high-level intuition about the modifications. For any fractional assignment matrix $F$, for any subset $I$ and paper $p$, the expected number of reviewers from subset $I$ assigned to paper $p$ is $\sum_{r \in I} F_{rp}$. This is equal to the initial load from subset $I$ on paper $p$ in the algorithm (that is, the sum of the flow on all edges from reviewers in subset $I$ to paper $p$). Note that at the algorithm's conclusion, when all edges are integral, the load from subset $I$ on paper $p$ is equal to the number of reviewers from subset $I$ assigned to paper $p$. Therefore, if the fractional assignment $F$ is such that the initial expected number of reviewers from subset $I$ assigned to paper $p$ is no greater than 1 (as stated in part (i) of Lemma 1), we want to keep the load from subset $I$ on paper $p$ close to its initial value so that the final number of reviewers from subset $I$ assigned to paper $p$ is also no greater than 1. With this reasoning, we modify the algorithm from Theorem 1 so that in each iteration, it ensures that the total load on each paper from each subset is unchanged if originally integral and is never moved past the closest integer in either direction if originally fractional. The time complexity of this modified algorithm is identical to that of the original algorithm from Theorem 1.

## 5.2 Finding the Optimal Partition-Constrained Fractional Assignment

Lemma 1 provides necessary and sufficient conditions for the fractional assignment matrices for which it is possible to prevent all pairs of same-subset reviewers from being assigned to the same paper. Therefore, to find an optimal fractional assignment with this property, we construct a new LP, $\mathcal{LP}2$, by adding $md$ constraints to $\mathcal{LP}1$:

$$\sum_{r \in I} F_{rp} \leq 1 \quad \forall I \in \{I_1, \ldots, I_m\}, \forall p \in \mathcal{P}. \tag{6}$$

The solution to $\mathcal{LP}2$ when paired with the sampling algorithm from Section 5.1 never assigns two reviewers from the same subset to the same paper. Furthermore, since any fractional assignment $F$ not obeying Constraint (6) will have a strictly positive probability of assigning two reviewers from the same subset to the same paper, $\mathcal{LP}2$ finds the optimal fractional assignment with this guarantee. This completes the algorithm for the Partition-Constrained Problem.

Additionally, Corollary 1 shows that the sampling algorithm from Section 5.1 is optimal in the expected number of same-subset reviewer pairs, for any fractional assignment. If the guarantee of entirely preventing same-subset reviewer pairs is not strictly required, Constraint (6) in $\mathcal{LP}2$ can be loosened (constraining the subset loads to a higher value) without removing it entirely. For the resulting fractional assignment $F$, the sampling algorithm from Section 5.1 still minimizes the expected number of pairs of reviewers from any subset on any paper, as compared to any other sampling algorithm implementing $F$. Since the subset loads are still constrained, the expected number of same-subset reviewer pairs will be lower than in the solution to the Pairwise-Constrained Problem (at the cost of some similarity). We examine this tradeoff experimentally in Section 6.

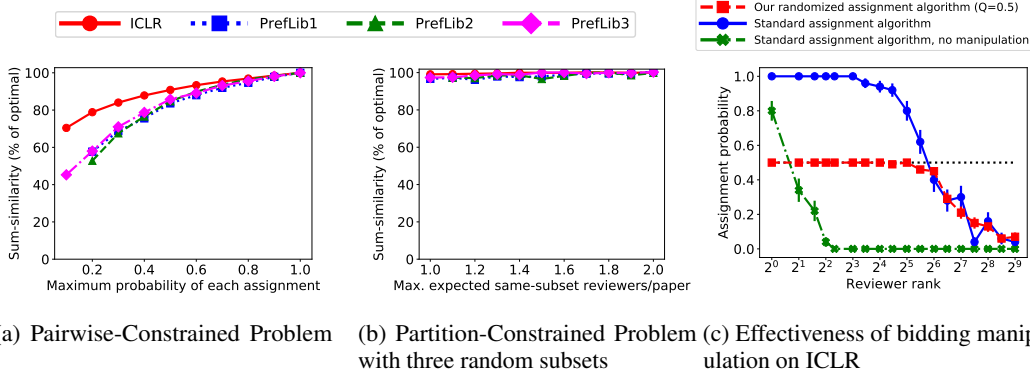

Figure 1: Experimental results on four conference datasets.

## 6 Experiments

We test our algorithms on several real-world datasets. The first real-world dataset is a similarity matrix recreated from ICLR 2018 data in [35]; this dataset has $n = 2435$ reviewers and $d = 911$ papers. We also run experiments on similarity matrices created from reviewer bid data for three AI conferences from PrefLib dataset MD-00002 [47], with sizes $(n = 31, d = 54)$, $(n = 24, d = 52)$, and $(n = 146, d = 176)$ respectively. For all three PrefLib datasets, we transformed "yes," "maybe," and "no response" bids into similarities of 4, 2, and 1 respectively, as is often done in practice [27]. As done in [35], we set loads $k = 6$ and $\ell = 3$ for all datasets since these are common loads for computer science conferences (except on the PrefLib2 dataset, for which we set $k = 7$ for feasibility).

We run all experiments on a computer with 8 cores and 16 GB of RAM, running Ubuntu 18.04 and using Gurobi 9.0.2 [48] to solve the LPs. Our algorithm for the Pairwise-Constrained Problem takes an average of 41 seconds to complete on ICLR; our algorithm for the Partition-Constrained Problem takes an average of 45 seconds. As expected, the running time is dominated by the LP solving.

### 6.1 Quality of Resulting Assignments

We first study our algorithm for the Pairwise-Constrained Problem, as described in Section 4. In this setting, program chairs must make a tradeoff between the quality of the output assignments and guarding against malicious reviewers or reviewer de-anonymization by setting the values of the maximum-probability matrix $Q$. We investigate this tradeoff on real datasets. All results in this section are averaged over 10 trials with error bars plotted representing the standard error of the mean, although they are sometimes not visible since the variance is very low.

In Figure 1a, we set all entries of the maximum-probability-matrix $Q$ equal to the same constant value $q_0$ (varied on the x-axis), and observe how the sum-similarity value of the assignment computed via our algorithm from Section 4 changes as $q_0$ increases from 0.1 to 1 with an interval of 0.1. We report the sum-similarity as a percentage of the unconstrained optimal solution's objective (which does not address the aforementioned challenges). We see that our algorithm trades off the maximum probability of an assignment gracefully against the sum-similarity on all datasets. For instance, with $q_0 = 0.5$, our algorithm achieves $90.8\%$ of the optimal objective value on the ICLR dataset. In practice, this would allow program chairs to limit the chance that any malicious reviewer is assigned to their desired paper to $50\%$ without suffering a significant loss of assignment quality.

We next test our algorithm for the Partition-Constrained Problem discussed in Section 5. In this algorithm, program chairs can navigate an additional tradeoff between the number of same-subset reviewers assigned to the same paper and the assignment quality; we investigate this tradeoff here. On ICLR, we fix $q_0 = 0.5$ and randomly assign reviewers to subsets of size 15 (representing academic institutions, for example), using this as our partition of $\mathcal{R}$ (since the dataset does not include any reviewer information). Our algorithm is able to achieve $100\%$ of the optimal objective for the Pairwise-Constrained Problem with $q_0 = 0.5$ while preventing any pairs of reviewers from the same subset from being assigned to the same paper.

Since our algorithm achieves the full possible objective in this setting, we now run experiments with a considerably more restrictive partition constraint. In Figure 1b, we show an extreme case where we randomly assign reviewers to 3 subsets of equal size, again fixing $q_0 = 0.5$. We then gradually loosen the constraints on the expected number of same-subset reviewers assigned to the same paper by increasing the constant in Constraint (6) from 1 to 2 in increments of 0.1, shown on the x-axis. We plot the sum-similarity objective of the resulting assignment, expressed as a percentage of the optimal non-partition-constrained solution's objective (i.e., the solution to the Pairwise-Constrained Problem with $q_0 = 0.5$). Even in this extremely constrained case, we still achieve 99.1% of the non-partition-constrained objective while entirely preventing same-subset reviewer pairs on ICLR.

In Appendix G, we present results for additional experiments on synthetic similarities.We also run experiments for a fairness objective, which we present in Appendix B.

## 6.2 Effectiveness at Preventing Manipulation

We now describe experiments evaluating the effectiveness of our algorithm at preventing manipulation on the ICLR dataset against a simulated reviewer bidding model. We assume that there is one malicious reviewer, who is attempting to maximize their chances of being assigned to a target paper solely through bidding (and not through other means). Since the ICLR similarities are reconstructed purely from the text similarity with each reviewers' past work and do not contain any bidding, we supplement them with synthetic bids. The malicious reviewer bids positively on their target paper and negatively on all other papers, while the other (honest) reviewers bid according to a simple randomized model constructed to match characteristics of the bidding observed in NeurIPS 2016 [27]. The function used to compute final similarities is based on the similarity function used in NeurIPS 2016 [27]. Details of the bidding are given in Appendix H, along with additional results.

In Figure 1c, we choose a target paper uniformly at random, and choose the malicious reviewer to be the reviewer with the $x^{\text{th}}$ highest pre-bid similarity with that paper (varying $x$ on the x-axis, log-scaled). We then have all reviewers bid and compute the assignment with either the standard deterministic assignment algorithm described in Section 3 or our randomized assignment algorithm for the Pairwise-Constrained Problem, setting all entries of $Q$ to 0.5. We then observe the probability that the malicious reviewer is assigned to the target paper. We report on the y-axis the average over 50 choices of target paper, giving an overall success rate for the manipulation under a uniform choice of papers (with error bars representing the standard error of the mean). For comparison, we also plot the case where only the honest reviewers bid and the malicious reviewer does not bid.

We see that under the standard assignment algorithm, the manipulation always succeeds for reviewers with high pre-bid similarity with the paper. In contrast, when the malicious reviewer does not bid, their assignment probability is much lower, especially for reviewers ranked in the top 4. This indicates that manipulation from reviewers in the same subject area as the paper is very powerful in standard assignment algorithms, potentially compromising the integrity of the assignment. However, our algorithm always limits the probability of successful manipulation to the desired level of 0.5.

## 7 Discussion

We have presented here a framework and a set of algorithms for addressing three challenges of practical importance to the peer review process: untruthful favorable reviews, torpedo reviewing, and reviewer de-anonymization on the release of assignment data. By design, our algorithms are quite flexible to the needs of the program chairs, depending on which challenges they are most concerned with addressing. Our empirical evaluations demonstrate some of the tradeoffs that can be made between total similarity and maximum probability of each paper-reviewer pair or number of reviewers from the same subset on the same paper. The exact parameters of the algorithm can be set based on how the program chairs weigh the relative importance of each of these factors.

This work leads to a number of open problems of interest. First, since the Triplet-Constrained Problem is NP-hard, we considered one special structure—the Partition-Constrained Problem—of practical relevance. A direction for future research is to find additional special cases under which optimizing over constraints on the probabilities of reviewer-pair-to-paper assignments is feasible. Additionally, this work does not address the problem of reviewers colluding with each other to give dishonest favorable reviews after being assigned to each others' papers; we leave this issue for future work.

## Broader Impact

We believe that our work can have a significant positive impact on the peer review processes of major conferences and journals. By mitigating our identified challenges, both program chairs and authors should benefit from the increased truthfulness of reviews. In addition, by allowing for paper assignment data and algorithms to be publicly released, this work can increase the transparency of the reviewing process and provide data for future research in this area. As we show in the experiments section, there is a tradeoff between the total similarity of the assignment and the mitigation of the challenges, so use of these algorithms will likely result in slightly poorer paper assignments in terms of reviewer fit. However, the program chairs are free to navigate this tradeoff as they see appropriate. An organizer's poor choice of parameters could result in a poor paper assignment and hurt the quality of the published research, but this is the kind of choice that program chairs must always make with respect to the peer review process.

By allowing program chairs to set different maximum probabilities for different paper-reviewer pairs, we do allow human biases to creep into the paper assignment process. To remedy this, we suggest setting the maximum probabilities in a principled way (such as according to a formula) that does not discriminate against papers or reviewers unfairly. Even if an individual paper-reviewer pair is unjustly suspected of being a bad match and the probability of that match is set low as a result, this alone is unlikely to negatively impact either the paper or the reviewer; this would be less problematic than if, for example, the paper is rejected because the program chairs are uncertain whether the reviews are trustworthy.

## Acknowledgments

The research of Steven Jecmen and Nihar Shah was supported in part by NSF CAREER 1942124. The research of Steven Jecmen and Fei Fang was supported in part by NSF Award IIS-1850477. The research of Hanrui Zhang and Vincent Conitzer was supported in part by NSF Award IIS-1814056.

## Footnotes

[1]`https://github.com/theryanl/mitigating_manipulation_via_randomized_reviewer_assignment/`

[2]For ease of exposition, we assume that all reviewer and paper loads are equal. In practice, program chairs may want to set different loads for different reviewers or papers; all of our algorithms and guarantees still hold for this case (as does our code).

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
