[Supplementary Material]

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

# Appendices

## A    Additional Related Literature

We here review some additional related literature.

A concurrent work [49] considers a different set of problems in releasing data in peer review while preserving reviewer anonymity. The data to be released here are some function of the scores and the reviewer-assignment, whereas we look to release the similarities and the assignment code. Moreover, the approach and techniques in [49] are markedly different—they consider post-processing the data for release using techniques such as differential privacy, whereas we consider randomizing the assignment for plausible deniability.

Outside of peer review, there is a line of prior work focused on detecting and mitigating manipulation in online reviews (such as those on Yelp or Amazon) [50–57]. These works typically make assumptions that are not applicable or require data that is not available in our setting. Several of these works analyze the graph of user reviews in order to detect fraudulent reviewers [50]. For example, [51] detects fraud from the review graph by assuming that products are trying to maximize the number of positive reviews they get, whereas in our setting it is important to mitigate the effectiveness of just a single author-paper collusion since the number of reviews per paper is fixed and small. Additionally, some of these works [52–54] are based on estimating the "true quality" of each item from the review graph, which is not possible in peer review since paper evaluations are subjective. Another direction of prior work uses machine learning to detect malicious behavior. Some research [55] detects fraud from reviewer statistics such as rating variance or number of ratings, but in the peer review setting, there is so little data for each reviewer that such features would be highly noisy or completely uninformative. Other works [56] attempt to detect malicious reviews from the review text, but in our setting, malicious reviews may not differ stylistically from genuine reviews. Finally, some work in recommender systems focuses on making product recommendations resistant to malicious reviews [57]; however, in peer review, the paper acceptance process must be done by hand since it must take into account reviewer opinions, arguments, and the reviewer discussion.

## B    Stochastic Fairness Objective

An alternate objective to the sum-similarity objective has been studied in past work [33, 34], aiming to improve the fairness of the assignment with respect to the papers. Rather than maximizing the sum-similarity across all papers, this objective maximizes the minimum total similarity assigned to any paper:

$$\underset{M \in \mathbb{R}^{n \times d}}{\arg\max} \quad \min_{p \in \mathcal{P}} \sum_{r \in \mathcal{R}} S_{rp} M_{rp}$$

$$\text{subject to} \quad M_{rp} \in \{0, 1\} \qquad \forall r \in \mathcal{R}, \forall p \in \mathcal{P}$$

$$\sum_{p \in \mathcal{P}} M_{rp} \leq k \qquad \forall r \in \mathcal{R}$$

$$\sum_{r \in \mathcal{R}} M_{rp} = \ell \qquad \forall p \in \mathcal{P}.$$

Due to the minimum in the objective, this problem is NP-hard [58]; the paper [34] presents an algorithm to find an approximate solution.

In our setting of randomized assignments, we consider an analogous fairness objective, which we call the stochastic fairness objective: $\min_{p \in \mathcal{P}} \mathbb{E}\left[\sum_{r \in \mathcal{R}} S_{rp} M_{rp}\right]$. The problem involving this objective is defined as follows.

**Definition 4** (Fair Pairwise-Constrained Problem). *The input to the problem is a similarity matrix $S$ and a matrix $Q \in [0, 1]^{n \times d}$. The goal is to find a randomized assignment of papers to reviewers that maximizes $\min_{p \in \mathcal{P}} \mathbb{E}\left[\sum_{r \in \mathcal{R}} S_{rp} M_{rp}\right]$ subject to the constraints that $\mathbb{P}[M_{rp} = 1] \leq Q_{rp}, \forall r \in \mathcal{R}, p \in \mathcal{P}$.*

This problem definition is identical to that of the Pairwise-Constrained Problem (Definition 1), with the exception that the objective to maximize is now the stochastic fairness objective rather

Figure 2: Experimental results for the Fair Pairwise-Constrained Problem.

than the sum-similarity. Note that this objective is not equal to the "expected fairness" (i.e., $\mathbb{E}\left[\min_{p \in \mathcal{P}} \sum_{r \in \mathcal{R}} S_{rp} M_{rp}\right]$), but by Jensen's inequality it is an upper bound on the expected fairness.

Fortunately, this problem is solvable efficiently, as the following theorem states.

**Theorem 4.** *There exists an algorithm which returns an optimal solution to the Fair Pairwise-Constrained Problem in poly(n, d) time.*

We now present our algorithm for solving the Fair Pairwise-Constrained Problem, thereby proving the theorem. It proceeds in a similar manner as the algorithm for the Pairwise-Constrained Problem presented in Section 4.

The algorithm first finds an optimal fractional assignment matrix, since the stochastic fairness objective depends only on the marginal probabilities in the fractional assignment matrix. The optimal fractional assignment is found by the following LP, which we call $\mathcal{LP}3$:

$$\underset{F \in \mathbb{R}^{n \times d}, x \in \mathbb{R}}{\arg \max} \quad x \tag{7}$$

$$\text{subject to} \quad 0 \leq F_{rp} \leq 1 \qquad \forall r \in \mathcal{R}, \forall p \in \mathcal{P} \tag{8}$$

$$\sum_{p \in \mathcal{P}} F_{rp} \leq k \qquad \forall r \in \mathcal{R} \tag{9}$$

$$\sum_{r \in \mathcal{R}} F_{rp} = \ell \qquad \forall p \in \mathcal{P} \tag{10}$$

$$F_{rp} \leq Q_{rp} \qquad \forall r \in \mathcal{R}, \forall p \in \mathcal{P} \tag{11}$$

$$x \leq \sum_{r \in \mathcal{R}} S_{rp} F_{rp} \qquad \forall p \in \mathcal{P}. \tag{12}$$

For any $F$, the optimal value of $x$ is always $\min_{p \in \mathcal{P}} \sum_{r \in \mathcal{R}} S_{rp} F_{rp}$, the stochastic fairness of $F$. For a fixed $x$, the feasible region of $F$ in $\mathcal{LP}3$ is exactly the space of fractional assignment matrices with stochastic fairness no less than $x$. Therefore, $\mathcal{LP}3$ will find an optimal fractional assignment matrix for the stochastic fairness objective.

Once an optimal fractional assignment matrix has been found, it only remains to sample a deterministic assignment from it. This is done with the sampling algorithm described in Section 4.2, just as in the Pairwise-Constrained Problem.

We now present some empirical results for this algorithm on the four conference datasets described in Section 6. We set all entries of $Q$ equal to the same constant value $q_0$ (varied on the x-axis), and observe how the stochastic fairness objective of the assignment changes as $q_0$ increases from $0.1$ to $1$ with an interval of $0.1$. Since the expectation is inside a minimum in the objective, the objective cannot be estimated without bias by averaging together the stochastic fairness of sampled deterministic assignments. Due to this difficulty, we plot the exact objective of our randomized assignment (i.e., the optimal objective value of $\mathcal{LP}3$) rather than averaging over multiple samples, and report the objective as a percentage of the unconstrained optimal solution's objective (that is, the

algorithm's solution when $q_0 = 1$). As Figure 2 shows, our algorithm finds a randomized assignment achieving 92.7% of the optimal fairness objective on the ICLR dataset when $q_0 = 0.5$.

## C   Bad-Assignment Probability Problem Variants

An input to both the Pairwise-Constrained Problem (Definition 1) and the Partition-Constrained Problem (Definition 3) is the matrix $Q$, where $Q_{rp}$ denotes the maximum probability with which reviewer $r$ should be assigned to paper $p$. In practice, program chairs can set the values in this matrix based on their own beliefs about each reviewer-paper pair. However, it may be difficult for program chairs to translate their beliefs about the risk of assigning any reviewer-paper pair into appropriate values for $Q$. In this appendix, we define alternate versions of these problems that allow the program chairs to codify their beliefs in a different way.

Define the assignment of reviewer $r$ to paper $p$ as "bad" if reviewer $r$ intends to untruthfully review paper $p$ (either because they intend to give a dishonest favorable review or because they intend to torpedo-review). Further define a matrix $W \in [0,1]^{n \times d}$ of bad-assignment probabilities, where $W_{rp}$ represents the probability that the assignment of reviewer $r$ to paper $p$ would be a bad assignment; we assume that the events of each reviewer-paper assignment being bad are all independent of each other. The "true value" of $W$ may not be known, but it can be set based on the program chairs' beliefs about the reviewers and authors or potentially estimated based on some data from prior conferences. The problem variants we present in the following subsections make use of these bad-assignment probabilities.

We first consider the problem of limiting the probabilities of bad reviewer-paper assignments. We then consider the problem of limiting the probabilities that bad pairs of reviewers are assigned to the same paper.

### C.1   Handling Bad Reviewer-Paper Assignments

We define an alternate version of the Pairwise-Constrained Problem using the bad-assignment probabilities:

**Definition 5** (Bad-Assignment Probability Pairwise-Constrained Problem). *The input to the problem is a similarity matrix $S$, a matrix $W \in [0,1]^{n \times d}$ of bad-assignment probabilities, and a value $\lambda \in [0,1]$. The goal is to find a randomized assignment of papers to reviewers that maximizes $\mathbb{E}\left[\sum_{r \in \mathcal{R}} \sum_{p \in \mathcal{P}} S_{rp} M_{rp}\right]$ subject to the constraints that $W_{rp}\mathbb{P}[M_{rp} = 1] \leq \lambda, \forall r \in \mathcal{R}, p \in \mathcal{P}$.*

$W_{rp}\mathbb{P}[M_{rp} = 1]$ is exactly the probability that both (i) reviewer $r$ is assigned to paper $p$ and (ii) this assignment is bad, so the constraints in the problem limit this at $\lambda$ for all $r \in \mathcal{R}$ and $p \in \mathcal{P}$. This version of the Pairwise-Constrained Problem may be useful in practice if program chairs find it easier to set the values of $W$ than they would for $Q$.

We now show how to solve the Bad-Assignment Probability Pairwise-Constrained Problem, by translating it to the original Pairwise-Constrained Problem. Suppose that we have access to the matrix $F$ of marginal assignment probabilities that occur under some randomized assignment. The randomized assignment obeys our constraints if and only if $F_{rp}W_{rp} \leq \lambda, \forall r \in \mathcal{R}, p \in \mathcal{P}$. This observation leads to the following method of solving the Bad-Assignment Probability Pairwise-Constrained Problem:

- Transform the given instance of the Bad-Assignment Probability Pairwise-Constrained Problem into an instance of the Pairwise-Constrained Problem by constructing a matrix of maximum probabilities $Q$ where
$$Q_{rp} = \min \{\lambda/W_{rp}, 1\} \qquad \forall r \in \mathcal{R}, p \in \mathcal{P}.$$

- Solve the Pairwise-Constrained Problem using the algorithm from Theorem 1, described in Section 4.

### C.2   Handling Bad Pairs of Reviewers

Here, we first present an alternative version of the Partition-Constrained Problem and show how to solve it. We then present a different approach to handling the issue of bad reviewer pairs.

### C.2.1 Constraints on Disjoint Reviewer Sets

In the same way as done above for the Pairwise-Constrained Problem, we define an alternate version of the Partition-Constrained Problem:

**Definition 6** (Bad-Assignment Probability Partition-Constrained Problem). *The input to the problem is a similarity matrix $S$, a matrix $W \in [0,1]^{n \times d}$ of bad-assignment probabilities, a value $\lambda \in [0,1]$, and a partition of the reviewer set into subsets $I_1, \dots, I_m \subseteq \mathcal{R}$. The goal is to find a randomized assignment of papers to reviewers that maximizes $\mathbb{E}\left[\sum_{r \in \mathcal{R}} \sum_{p \in \mathcal{P}} S_{rp} M_{rp}\right]$ subject to the constraints that $W_{rp}\mathbb{P}[M_{rp} = 1] \leq \lambda, \forall r \in \mathcal{R}, p \in \mathcal{P}$ and $\mathbb{P}[M_{ap} = 1 \wedge M_{bp} = 1] = 0, \forall a, b \in I_i, \forall i \in [m]$.*

Just as for the Bad-Assignment Probability Pairwise-Constrained Problem, we solve this problem by first transforming an instance of this problem into an equivalent instance of the Partition-Constrained Problem, done by constructing a matrix of maximum probabilities $Q$ where $Q_{rp} = \min(\lambda/W_{rp}, 1), \forall r \in \mathcal{R}, p \in \mathcal{P}$. We then solve this instance using the algorithm in Section 5.

### C.2.2 Constraints on the Expected Number of Bad Reviewers

The Bad-Assignment Probability Partition-Constrained Problem requires a partition of the reviewer set and prevents pairs of reviewers from being assigned to the same paper if they are in the same subset of this partition. Alternatively, one may want to prevent pairs of reviewers from being assigned to the same paper based on whether $W$ indicates that they are both likely to be bad assignments on this paper, rather than based on some partition of the reviewer set. In this way, we now present an alternative approach to handling the issue of bad reviewer pairs, which does not require a partition of the reviewer set. Rather than explicitly constraining the probabilities of certain same-subset reviewer-reviewer-paper triples as in the Bad-Assignment Partition-Constrained Problem, we limit the *expected* number of bad reviewers on each paper.

The following problem states this goal:

**Definition 7** (Bad-Assignment Probability Expectation-Constrained Problem). *The input to the problem is a similarity matrix $S$, a matrix $W \in [0,1]^{n \times d}$ of bad-assignment probabilities, a value $\lambda \in [0,1]$, and a value $\mu \in \mathbb{R}$. The goal is to find a randomized assignment of papers to reviewers that maximizes $\mathbb{E}\left[\sum_{r \in \mathcal{R}} \sum_{p \in \mathcal{P}} S_{rp} M_{rp}\right]$ subject to the constraints that $W_{rp}\mathbb{P}[M_{rp} = 1] \leq \lambda, \forall r \in \mathcal{R}, p \in \mathcal{P}$ and $\sum_{r \in \mathcal{R}} W_{rp}\mathbb{E}[M_{rp}] \leq \mu, \forall p \in \mathcal{P}$.*

We now present the algorithm that optimally solves this problem. The following LP, $\mathcal{LP}4$, finds a fractional assignment with expected number of bad reviewers on each paper no greater than $\mu$:

$$\underset{F \in \mathbb{R}^{n \times d}}{\arg\max} \quad \sum_{p \in \mathcal{P}} \sum_{r \in \mathcal{R}} S_{rp} F_{rp} \tag{13}$$

$$\text{subject to} \quad 0 \leq F_{rp} \leq 1 \qquad\qquad \forall r \in \mathcal{R}, \forall p \in \mathcal{P} \tag{14}$$

$$\sum_{p \in \mathcal{P}} F_{rp} \leq k \qquad\qquad \forall r \in \mathcal{R} \tag{15}$$

$$\sum_{r \in \mathcal{R}} F_{rp} = \ell \qquad\qquad \forall p \in \mathcal{P} \tag{16}$$

$$F_{rp} W_{rp} \leq \lambda \qquad\qquad \forall r \in \mathcal{R}, \forall p \in \mathcal{P} \tag{17}$$

$$\sum_{r \in \mathcal{R}} F_{rp} W_{rp} \leq \mu \qquad\qquad \forall p \in \mathcal{P}. \tag{18}$$

Constraints (14-16) define the space of fractional assignment matrices, Constraint (17) ensures that the probability of each bad assignment occurring is limited at $\lambda$, and Constraint (18) ensures that the expected number of bad reviewer-paper assignments for each paper is at most $\mu$. Therefore, $\mathcal{LP}4$ finds the optimal fractional assignment for the Bad-Assignment Probability Expectation-Constrained Problem. This fractional assignment can then be sampled from using the sampling algorithm in Section 4.2.

The above approach to controlling bad reviewer pairs is not directly comparable to the approach taken earlier when solving the Bad-Assignment Probability Partition-Constrained Problem. The

Bad-Assignment Probability Expectation-Constrained Problem indirectly restricts pairs of reviewers from being assigned to the same paper based on whether $W$ indicates that they are both likely to be bad assignments on that paper, instead of based on a partition of the reviewer set. This could be advantageous if the sets of likely-bad reviewers for each paper (as given by the probabilities in $W$) are not expressed well by any partition of the reviewer set. However, handling suspicious reviewer pairs through constraining the expected number of bad reviewers per paper is weaker than directly constraining the probabilities of certain reviewer-reviewer-paper triples (as in the Bad-Assignment Probability Partition-Constrained Problem). First, it provides a guarantee only in expectation, and does not guarantee anything about the probabilities of the events we wish to avoid (that is, bad reviewer pairs being assigned to a paper). In addition, we here are assuming that the event of paper $p$ and reviewer $r$ being a bad assignment is independent of this event for all other reviewer-paper pairs; so, this method cannot address the issue of associations between reviewers, such as their presence at the same academic institution.

# D  Algorithms for the Pairwise-Constrained Problem

In Section 4, we briefly sketched the sampling algorithm that realizes Theorem 1, a simplified version of the algorithm from [43]. Here, we first present this sampling algorithm in detail, thus solving the Pairwise-Constrained Problem (Definition 1). We then present a supplementary algorithm to compute the full distribution over deterministic assignments, which [43] does not. Knowing the full distribution may be useful in order to compute other properties of the randomized assignment not calculable from $F$ directly.

## D.1  Sampling Algorithm

Recall that $\mathcal{LP}1$ finds the optimal marginal assignment probabilities $F$ for the Pairwise-Constrained Problem. In this subsection, we present the sampling algorithm that implements these marginal probabilities as a randomization over deterministic paper assignments. The sampling algorithm proceeds as follows. We begin by constructing a directed graph $G = (V, E)$ for our problem, along with a capacity function $h : E \to \mathbb{Z}$ (Lines 1-3). First, construct one vertex for each reviewer, one vertex for each paper, and source and destination vertices $s, t$. Add an edge from the source vertex to each reviewer's vertex with capacity $k$. Add an edge from each paper's vertex to the destination vertex with capacity $\ell$. Finally, add an edge from each reviewer to each paper with capacity 1. We also construct a flow function $f : E \to \mathbb{R}$, which obeys the flow conservation constraints $\sum_{e \in E \cap (V \times \{v\})} f(e) = \sum_{e \in E \cap (\{v\} \times V)} f(e), \forall v \in V \setminus \{s, t\}$ and the capacity constraints $f(e) \leq h(e), \forall e \in E$ (Line 4). A (possibly fractional) assignment $F$ can be represented as a flow on this graph, where the flow from reviewer $i$ to paper $j$ corresponds to the probability reviewer $i$ is assigned to paper $j$ and the other flows are set uniquely by flow conservation. Due to the load constraints on assignments, the flows on the edges from the papers to the destination must be equal to those edges' capacities and the flows on the edges from the source to the reviewers must be less than or equal to the capacities.

The algorithm then proceeds in an iterative manner, modifying the flow function $f$ on each iteration. On each iteration, we first check if there exists a "fractional edge," an edge with non-integral flow. If no such edge exists, our current assignment is integral and so we can stop iterating. If there does exist a fractional edge, we then find an arbitrary cycle of fractional edges, ignoring direction (Line 6); this can be done by starting at any fractional edge and walking along fractional edges until a previously-visited vertex is returned to. On finding a cycle, we randomly modify the flow on each of the edges in the cycle in order to guarantee that at least one of the flows becomes integral.

In what follows, we first prove that such a cycle of fractional edges can always be found. We then show how to modify the flows in order to guarantee the implementation of the marginal assignment probabilities.

We now show that a directionless cycle of fractional edges must exist whenever one fractional edge exists. Initially, by the properties of $F$, the total flow on each edge going into vertex $t$ is integral; further, the algorithm only ever changes the flow on edges with non-integral flow. Therefore, the total flow going into $t$ is always integral. By flow conservation, the total flow leaving $s$ is also always integral. So, if there is a fractional edge adjacent to $s$, there must also be another fractional edge adjacent to $s$. As already stated, there are no fractional edges adjacent to $t$. Finally, for each vertex

---

**Algorithm 1** Sampling algorithm for the Pairwise-Constrained Problem.

---

**Input:** Fractional assignment matrix $F$, reviewer set $\mathcal{R}$, paper set $\mathcal{P}$
**Ouput:** Deterministic assignment matrix $M$
**Algorithm:**

1: Construct vertex set $V \leftarrow \mathcal{R} \cup \mathcal{P} \cup \{s\} \cup \{t\}$
2: Construct directed edge set $E \leftarrow \{(r,p)|\forall r \in \mathcal{R}, p \in \mathcal{P}\} \cup \{(s,r)|\forall r \in \mathcal{R}\} \cup \{(p,t)|\forall p \in \mathcal{P}\}$
3: Construct capacity function $h : E \rightarrow \mathbb{Z}$ as $h(e) \leftarrow \begin{cases} 1 & \text{if } e \in \mathcal{R} \times \mathcal{P} \\ k & \text{if } e \in \{s\} \times \mathcal{R} \\ \ell & \text{if } e \in \mathcal{P} \times \{t\} \end{cases}$
4: Construct initial flow function $f : E \rightarrow \mathbb{R}$ as $f(e) \leftarrow \begin{cases} F_{rp} & \text{if } e = (r,p) \in \mathcal{R} \times \mathcal{P} \\ \sum_{p \in \mathcal{P}} F_{rp} & \text{if } e = (s,r) \in \{s\} \times \mathcal{R} \\ \sum_{r \in \mathcal{R}} F_{rp} & \text{if } e = (p,t) \in \mathcal{P} \times \{t\} \end{cases}$
5: **while** $\exists e \in E$ such that $f(e) \notin \mathbb{Z}$ **do**
6:    Find a cycle of edges (ignoring direction) $C = \{e_1, \ldots, e_k\}$ such that $f(e_i) \notin \mathbb{Z}, \forall i \in [k]$
7:    $A \leftarrow \{e \in C | \ e \text{ is directed in the same direction as } e_1 \text{ along the cycle}\}$
8:    $B \leftarrow C \setminus A$
9:    $\alpha \leftarrow \min\left(\min_{e \in A} f(e), \min_{e \in B} h(e) - f(e)\right)$
10:    **for** $e \in A$ **do**
11:        $f_1(e) \leftarrow f(e) - \alpha$
12:    **end for**
13:    **for** $e \in B$ **do**
14:        $f_1(e) \leftarrow f(e) + \alpha$
15:    **end for**
16:    $\beta \leftarrow \min\left(\min_{e \in A} h(e) - f(e), \min_{e \in B} f(e)\right)$
17:    **for** $e \in A$ **do**
18:        $f_2(e) \leftarrow f(e) + \beta$
19:    **end for**
20:    **for** $e \in B$ **do**
21:        $f_2(e) \leftarrow f(e) - \beta$
22:    **end for**
23:    $\gamma \leftarrow \frac{\beta}{\alpha + \beta}$
24:    With probability $\gamma$, $f \leftarrow f_1$; else $f \leftarrow f_2$
25: **end while**
26: $M_{rp} = f((r,p)), \forall (r,p) \in \mathcal{R} \times \mathcal{P}$

---

$v \in V \setminus \{s,t\}$, by flow conservation, there can never be only one fractional edge adjacent to $v$. Therefore, every vertex that is adjacent to a fractional edge must also be adjacent to another fractional edge. This proves that a directionless cycle of fractional edges must exist if one fractional edge exists.

We now show how to modify the flow on the edges in this cycle. We can keep pushing flow in some direction on this cycle (pushing negative flow if the edge is directed backwards) until some edge is at capacity or has 0 flow. Call this amount of additional flow $\alpha$, and the resulting flow $f_1$. We can do the same thing in the other direction on the cycle, calling the additional flow $\beta$ and the resulting flow $f_2$. Both $f_1$ and $f_2$ must have at least one more integral edge than $f$, since some edge is at capacity. Further, both $f_1$ and $f_2$ obey the flow conservation and capacity constraints. Defining $\gamma \leftarrow \frac{\beta}{\alpha + \beta}$, we set $f \leftarrow f_1$ with probability $\gamma$ and $f \leftarrow f_2$ with probability $1 - \gamma$ (Lines 23-24).

Once all edges are integral (after the final iteration), we construct the sampled deterministic assignment $M$ from the flow on the reviewer-paper edges (Line 26). Since $f$ obeys the capacity constraints on all edges, $M$ obeys the load constraints and so is in fact an assignment. Since on each iteration the initial flow $f$ satisfies $f(e) = \gamma f_1(e) + (1 - \gamma) f_2(e), \forall e \in E$, the expected final flow on each edge is always equal to the current flow on that edge. Since the expectation of a Bernoulli random variable is exactly the probability it equals one, each final reviewer-paper assignment $M_{rp}$ has been chosen with the desired marginal probabilities $F_{rp}$.

### D.2 Decomposition Algorithm

We here provide a decomposition algorithm to compute a full distribution over deterministic assignments for a given fractional assignment matrix (which the prior work [43] does not). For simplicity, we assume here that all reviewer loads are met with equality (that is, $\sum_{p \in \mathcal{P}} F_{rp} = k$ for all $r \in \mathcal{R}$); the extension to the case when reviewer loads are met with inequality is simple.

We first define certain concepts necessary for the algorithm. We then present a subroutine of the algorithm and prove its correctness. We then present the overall algorithm and prove its correctness. Finally, we analyze the time complexity of the algorithm.

**Preliminaries:** We define here three concepts used in the algorithm and its proof.

- A capacitated matching instance consists of a set of papers $\mathcal{P}$, a set of reviewers $\mathcal{R}$, and a capacity function $h : \mathcal{P} \cup \mathcal{R} \to \mathbb{Z}$. A solution to $(\mathcal{P}, \mathcal{R}, h)$ is a matrix $F \in [0,1]^{n \times d}$, where for any $p \in \mathcal{P}$,

$$\sum_{r \in \mathcal{R}} F_{rp} = h(p),$$

  and for any $r \in \mathcal{R}$,

$$\sum_{p \in \mathcal{P}} F_{rp} = h(r).$$

  The solution $F$ is integral if $F_{rp} \in \{0,1\}$ for all $p \in \mathcal{P}$ and $r \in \mathcal{R}$.

- For any $\mathcal{R}$ and $\mathcal{P}$, a maximum matching on a set $S \subseteq \mathcal{R} \times \mathcal{P}$ subject to capacities $h$ is a set $M \subseteq S$ such that $\sum_{r \in \mathcal{R}} \mathbb{I}[(r,p) \in M] \leq h(p), \forall p \in \mathcal{P}$ and $\sum_{p \in \mathcal{P}} \mathbb{I}[(r,p) \in M] \leq h(r), \forall r \in \mathcal{R}$, and $|M|$ is maximized.

- For any $\mathcal{R}$ and $\mathcal{P}$, a perfect matching on a set $S \subseteq \mathcal{R} \times \mathcal{P}$ subject to capacities $h$ is a maximum matching on $S$ subject to $h$ that additionally satisfies $\sum_{r \in \mathcal{R}} \mathbb{I}[(r,p) \in M] = h(p), \forall p \in \mathcal{P}$ and $\sum_{p \in \mathcal{P}} \mathbb{I}[(r,p) \in M] = h(r), \forall r \in \mathcal{R}$.

**Decomposition subroutine:** The following procedure, a subroutine of the overall algorithm, takes an instance $(\mathcal{P}, \mathcal{R}, h)$ and a solution to that instance $F$ as input, and outputs an integral solution $F_0$ to $(\mathcal{P}, \mathcal{R}, h)$ with weight $\alpha_0$ and a fractional solution $F'$ to $(\mathcal{P}, \mathcal{R}, h)$ with strictly fewer fractional entries than $F$. Moreover, $F$, $F_0$, $\alpha_0$, and $F'$ satisfy $F = \alpha_0 F_0 + (1 - \alpha_0) F'$.

1. Let $E \subseteq \mathcal{R} \times \mathcal{P}$ be $E = \{(r,p) \mid F_{rp} \in (0,1)\}$, and let $M_0 \subseteq \mathcal{R} \times \mathcal{P}$ be $M_0 = \{(r,p) \mid F_{rp} = 1\}$. With this, define capacity function $h'$ as, for any $p \in \mathcal{P}$,

$$h'(p) = h(p) - |\{(r,p) \mid r \in \mathcal{R}\} \cap M_0|$$

   and for any $r \in \mathcal{R}$,

$$h'(r) = h(r) - |\{(r,p) \mid p \in \mathcal{P}\} \cap M_0|.$$

2. Find a maximum matching $M \subseteq E$ on $E$ subject to capacity constraints $h'$.

3. Set $F_0$ as

$$(F_0)_{rp} = \mathbb{I}\left[(r,p) \in M \cup M_0\right], \forall r \in \mathcal{R}, p \in \mathcal{P}.$$

   Set $F'$ as

$$F'_{rp} = \frac{1}{(1 - \alpha_0)}(F_{rp} - \alpha_0 (F_0)_{rp}), \forall r \in \mathcal{R}, p \in \mathcal{P}.$$

   Set $\alpha_0 = \min(\{F_{rp} \mid (r,p) \in M\} \cup \{1 - F_{rp} \mid (r,p) \in E \setminus (M \cup M_0)\})$.

We prove the correctness of this subroutine in Lemma 3. Before we do, we restate a result from prior work [43] that we use in the proof, using our own notation.

**Lemma 2** ( [43, Thm. 1]). *For any $(\mathcal{P}, \mathcal{R}, h)$ and any solution $F$ to $(\mathcal{P}, \mathcal{R}, h)$, there exists some $z \in \mathbb{Z}$, integral solutions $\{F_1, \ldots, F_z\}$ to $(\mathcal{P}, \mathcal{R}, h)$, and $\alpha$ lying on the $z$-dimensional simplex, such that $F = \sum_{i=1}^{z} \alpha_i F_i$.*

Now, the following lemma proves the correctness of the subroutine.

**Lemma 3.** *The decomposition subroutine finds $F_0$, $\alpha_0$, and $F'$, such that (i) $F_0$ is an integral solution to $(\mathcal{P}, \mathcal{R}, h)$, (ii) $F'$ is a fractional solution to $(\mathcal{P}, \mathcal{R}, h)$, (iii) $F'$ has strictly fewer fractional entries than $F$, and (iv) $F = \alpha_0 F_0 + (1 - \alpha_0)F'$.*

*Proof.* We first consider (i). The key step is to show that the maximum matching $M$ found in step 2 is a perfect matching with respect to $h'$, or equivalently, to show there is a perfect matching on $E$ with respect to $h'$. Consider the capacitated matching instance $(\mathcal{P}, \mathcal{R}, h')$, and the solution $F''$ where

$$F''_{rp} = \begin{cases} F_{rp} & \text{if } F_{rp} < 1 \\ 0 & \text{otherwise.} \end{cases}$$

$F''$ is a solution to $(\mathcal{P}, \mathcal{R}, h')$ by the construction of $h'$. By Lemma 2, $F''$ is a convex combination of integral solutions to $(\mathcal{P}, \mathcal{R}, h')$. For some $z$, let $\{F_1, \ldots, F_z\}$ and $\alpha$ be such a decomposition of $F''$, where each $F_i$ is an integral solution to $(\mathcal{R}, \mathcal{P}, h')$ and $\alpha_i$ is its associated weight. For each $i \in [z]$, let $M_i \subseteq \mathcal{R} \times \mathcal{P}$ be the set of $(r, p)$ pairs where $(F_i)_{rp} = 1$. Since $F_i$ is a solution to $(\mathcal{R}, \mathcal{P}, h')$, $M_i$ is a perfect matching with respect to $h'$. By the definition of $F''$, $(r, p) \in E$ if and only if $F''_{rp} > 0$. Now since $F'' = \sum_{i=1}^z \alpha_i F_i$, $E = \bigcup_{i=1}^z M_i$. Since each $M_i$ is a perfect matching with respect to $h'$, $E$ contains a perfect matching with respect to $h'$ and so the maximum matching $M$ found is in fact a perfect matching with respect to $h'$. Therefore, $M \cup M_0$ is a perfect matching with respect to $h$ by the definition of $h'$. Therefore, $F_0$ is an integral solution to $(\mathcal{P}, \mathcal{R}, h)$.

For (ii), by the construction of $F'$, all capacity constraints hold with equality. We only need to show that $F'_{rp} \in [0, 1]$ for any $(r, p)$. Consider any $(r, p)$. There are 3 cases. If $(r, p) \in M_0$, then $F'_{rp} = 1$. If $(r, p) \notin M \cup M_0$, then the choice of $\alpha_0$ ensures that

$$F'_{rp} = \frac{1}{(1 - \alpha_0)} F_{rp} \leq \frac{1}{(1 - (1 - F_{rp}))} F_{rp} = 1$$

and

$$F'_{rp} = \frac{1}{(1 - \alpha_0)} F_{rp} \geq F_{rp} \geq 0.$$

If $(r, p) \in M$, the choice of $\alpha_0$ ensures that

$$F'_{rp} = \frac{1}{(1 - \alpha_0)}(F_{rp} - \alpha_0) \geq \frac{1}{(1 - \alpha_0)}(F_{rp} - F_{rp}) = 0$$

and

$$F'_{rp} = \frac{1}{(1 - \alpha_0)}(F_{rp} - \alpha_0) \leq F_{rp} \leq 1.$$

As a result, $F'$ is a solution to $(\mathcal{P}, \mathcal{R}, h)$.

For (iii), the choice of $\alpha_0$ ensures that at least one of the inequalities above achieves equality. That is, there exists $(r, p)$ where $F_{rp} \in (0, 1)$ such that $F'_{rp} \in \{0, 1\}$.

Finally, (iv) holds by the construction of $F_0$ and $F'$. $\qquad\square$

**Overall algorithm:** Using the above subroutine, the overall algorithm proceeds in the following recursive way. It takes as input a capacitated matching instance $(\mathcal{P}, \mathcal{R}, h)$ and a solution to that instance $F$. It outputs integral solutions $\{F_1, \ldots, F_z\}$ to $(\mathcal{P}, \mathcal{R}, h)$ and $\alpha$ lying on the $z$-dimensional simplex, such that $F = \sum_{i=1}^z \alpha_i F_i$.

1. If $F$ is integral, return solution $\{F\}$ and weight 1.

2. Otherwise, decompose $F$ into $F_0$ (with weight $\alpha_0$) and $F'$ using the above subroutine.

3. Recursively call this algorithm with $(\mathcal{P}, \mathcal{R}, h)$ and $F'$ as input, decomposing $F'$ into solutions $\{F_1, \ldots, F_z\}$ with weights $\alpha$.

4. Define $\beta = (1 - \alpha_0)\alpha$. Return the solutions $\{F_0, F_1, \ldots, F_z\}$ with weights $(\alpha_0, \beta_1, \ldots, \beta_z)$.

We now prove the correctness of this algorithm.

**Theorem 5.** *The decomposition algorithm correctly outputs integral solutions $\{F_1, \ldots, F_z\}$ to $(\mathcal{P}, \mathcal{R}, h)$ and $\alpha$ lying on the $z$-dimensional simplex, such that $F = \sum_{i=1}^z \alpha_i F_i$.*

*Proof.* We prove this statement by induction. If the algorithm returns in step 1, the theorem's statement holds. Now, assume that the theorem's statement holds for the decomposition returned by the recursive call to the algorithm in step 3, so that the following all hold: $\{F_1, \ldots, F_z\}$ are integral solutions to $(\mathcal{P}, \mathcal{R}, h)$, $\alpha$ lies on the $z$-dimensional simplex, and $F' = \sum_{i=1}^{z} \alpha_i F_i$. By Lemma 3, $F_0$ is an integral solution to $(\mathcal{P}, \mathcal{R}, h)$, so the $z + 1$ solutions returned in step 4 are integral solutions to $(\mathcal{P}, \mathcal{R}, h)$. Since $\alpha_0 \in [0, 1]$, $\beta \in [0, 1]^z$, and $\alpha_0 + \sum_{i=1}^{z} \beta_z = 1$, the weights returned in step 4 lie on the $z + 1$ dimensional simplex. Finally, by Lemma 3,

$$F = \alpha_0 F_0 + (1 - \alpha_0) F'$$

$$= \alpha_0 F_0 + (1 - \alpha_0) \sum_{i=1}^{z} \alpha_i F_i$$

$$= \alpha_0 F_0 + \sum_{i=1}^{z} \beta_i F_i.$$

Therefore, the theorem's statement holds for the output of the algorithm in step 4. By induction, this proves the desired statement. □

This decomposition algorithm can be used as part of the algorithm that solves the Pairwise-Constrained Problem, substituting for the sampling algorithm described in Section 4.2. It finds the full decomposition of the fractional assignment matrix into deterministic assignments rather than sampling a deterministic assignment. The capacity function $h$ used as the original input to the algorithm is defined as $h(r) = k, \forall r \in \mathcal{R}$ and $h(p) = \ell, \forall p \in \mathcal{P}$, and the input solution $F$ is exactly the fractional assignment matrix found as the solution to $\mathcal{LP}1$. The output integral solutions represent deterministic assignments, and the corresponding weights represent the probability with which each assignment should be chosen.

**Time complexity:** Since $F'$ has at least one fewer fractional entry than $F$, the recursive procedure has depth $O(dn)$ and therefore makes $O(dn)$ calls to the decomposition subroutine. In each call, the bottleneck is finding a maximum matching on $E$ subject to capacities $h$. This can be solved as a max-flow problem on a graph with $O(d + n)$ vertices and $O(dn)$ edges [59]. Using Dinic's algorithm [60], the computation of each matching takes $O(dn(d + n)^2)$ time, giving an overall time complexity of $O(d^2 n^2 (d + n)^2)$.

## E    Proof of Theorem 2 and a Corollary

We first define a decision variant of the Triplet-Constrained Problem, called "Arbitrary-Constraint Feasibility." An instance of this problem is defined by the paper and reviewer loads $\ell$ and $k$, and a 3-dimensional tensor $T \in [0, 1]^{n \times n \times d}$. For all $i, j \in \mathcal{R}, i \neq j$ and for all $p \in \mathcal{P}$, $T_{ijp}$ denotes the maximum probability that both reviewers $i$ and $j$ are assigned to paper $p$. The question is: does there exist a randomized assignment that obeys the constraints given by $T$? We next show that Arbitrary-Constraint Feasibility is NP-hard by a reduction from 3-Dimensional Matching.

An instance of 3-Dimensional Matching consists of three sets $X, Y, Z$ of size $s$, and a collection of tuples in $X \times Y \times Z$. It asks whether there exists a selection of $s$ tuples that includes each element of $X, Y$, and $Z$ at most once. This problem is known to be NP-complete [61].

Given such an instance of 3-Dimensional Matching, we construct an instance of Arbitrary-Constraint Feasibility. Set loads of $\ell = 2$ reviewers per paper and $k = 1$ paper per reviewer. Consider $|X| + |Y|$ reviewers (one for each element of $X \cup Y$) and $|Z|$ papers (one for each element of $Z$). Define the tensor $T$ to have $T_{ijp}$ equal to 1 if $(i, j, p)$ is one of the tuples, and 0 otherwise.

We now show that a 3-Dimensional Matching instance is a yes instance (that is, the answer to it is "yes") if and only if the corresponding Arbitrary-Constraint Feasibility instance is a yes instance, thus proving that solving Arbitrary-Constraint Feasibility in polynomial time would allow us to solve 3-Dimensional Matching in polynomial time. If there exists a feasible reviewer-paper assignment in the corresponding Arbitrary-Constraint Feasibility instance, then we would answer yes for the original 3-Dimensional Matching instance; otherwise, if there does not exist a feasible reviewer-paper assignment, then we would answer no for the original 3-Dimensional Matching instance.

If the 3-Dimensional Matching instance is a yes (that is, there exists a valid selection of $s$ tuples), then consider the paper assignment that assigns the corresponding reviewers and paper within each triple in the matching. Each paper has exactly 2 reviewers and each reviewer has exactly 1 paper, so this is a deterministic assignment. Since it includes only the triples in the matching instance, it obeys the probability constraints of $T$, so the Arbitrary-Constraint Feasibility instance is a yes.

If the 3-Dimensional Matching instance is a no, then all choices of $s$ tuples include some element of $X, Y,$ or $Z$ twice. If some element of $Z$ is chosen twice, then there must exist another element of $Z$ that is not included in any tuple. Therefore, any assignment of reviewer pairs to papers must either (a) include some reviewer-pair-to-paper assignment disallowed by $T$ (i.e., an assignment not in the collection of tuples), (b) make less than $s$ assignments of pairs to papers (and thus not assign to some paper), or (c) assign a reviewer twice or not assign some paper. So, no deterministic reviewer-paper assignment can meet the constraints of $T$. Now consider any randomized assignment, and select an arbitrary deterministic assignment in support of the randomized assignment. This deterministic assignment does not meet the constraints of $T$, so it must assign some reviewer $r$ to some paper $p$ that $T$ requires to have probability 0. Therefore, since this deterministic assignment is in support, the randomized assignment assigns reviewer $r$ to paper $p$ with non-zero probability, thereby violating the constraints of $T$. Therefore, no randomized assignment can meet the constraints of $T$. Therefore, the Arbitrary-Constraint Feasibility instance is a no. This proves that Arbitrary-Constraint Feasibility is NP-hard.

Since even telling if the feasible region of randomized assignments is non-empty is NP-hard, optimizing any objective over this region is also NP-hard. Therefore, the Triplet-Constrained Problem is NP-hard. This completes the proof.

Theorem 2 implies a more fundamental result about the feasible region of implementable reviewer-reviewer-paper probability tensors, that is, the tensors $G \in [0,1]^{n \times n \times d}$ where entry $G_{ijp}$ represents the marginal probability that both reviewers $i$ and $j$ are assigned to paper $p$ under some randomized assignment. We can represent any deterministic assignment by a 3-dimensional tensor $M \in \{0,1\}^{n \times n \times d}$ where $M_{ijp} = 1$ if and only if both reviewers $i$ and $j$ are assigned to paper $p$. Just as in the earlier case of fractional assignment matrices, the set of implementable probability tensors is a polytope with deterministic assignment tensors at the vertices (since any implementable probability tensor is a convex combination of deterministic assignment tensors). For fractional reviewer-paper assignment matrices, this polytope was defined by a small number ($O(dn)$) of linear inequalities, despite the fact that it has a large number of vertices (factorial in $d$ and $n$). However, this is no longer the case for reviewer-reviewer-paper probabilities.

**Corollary 2.** *The polytope of implementable reviewer-reviewer-paper probabilities is not expressible in a polynomial (in $n$ and $d$) number of linear inequality constraints (assuming $P \neq NP$).*

*Proof.* Suppose that the polytope of implementable reviewer-reviewer-paper probabilities could be expressed in a polynomial number of linear inequality constraints (with the reviewer-reviewer-paper probabilities as variables). An LP could then be constructed with these inequalities as well as the inequalities given by a tensor $T$ of maximum reviewer-reviewer-paper probabilities. Solving this LP with any linear objective would then find a feasible point, solving Arbitrary-Constraint Feasibility. Since LPs can be solved in time polynomial in the number of variables and constraints, this is a contradiction unless $P \neq NP$. $\qquad\square$

# F  Sampling Algorithm for the Partition-Constrained Problem and Related Proofs

In Section 5.1, we briefly sketched the sampling algorithm that realizes Lemma 1 and Corollary 1. Here, we first present this algorithm in detail. We then present proofs of the guarantees.

## F.1  Sampling Algorithm

The algorithm realizing Lemma 1 and Corollary 1 is obtained by changing three lines in Algorithm 1 (described in Appendix D), as follows:

- Line 6 is replaced with the subroutine in Algorithm 2.

**Algorithm 2** Loop-finding subroutine (replacing Line 6 in Algorithm 1).

---

1: Construct the set of undirected edges $E_U \leftarrow E \cup \{(v, u) \mid (u, v) \in E\}$
2: Construct the undirected flow function $f_U : E_U \rightarrow \mathbb{R}$ as $f_U((u, v)) \leftarrow$
$\begin{cases} f((u, v)) & \text{if } (u, v) \in E \\ f((v, u)) & \text{otherwise} \end{cases}$
3: Find arbitrary edge $(u, v) \in E$ such that $f((u, v)) \notin \mathbb{Z}$
4: $C \leftarrow \{(u, v)\}$
5: $D_1 \leftarrow \{\}, D_2 \leftarrow \{\}$
6: **while** $v$ has not previously been visited **do**
7: $\quad$ Visit $v$
8: $\quad$ **if** $u \in \mathcal{R}$ and $v \in \mathcal{P}$ **then**
9: $\quad\quad$ Set $I \in \{I_1, \ldots, I_m\}$ such that $u \in I$
10: $\quad\quad$ **if** $\exists w \in I \setminus \{u\}$ such that $(v, w) \in E_U$ and $f_U((v, w)) \notin \mathbb{Z}$ **then**
11: $\quad\quad\quad$ Find such a $w$
12: $\quad\quad$ **else**
13: $\quad\quad\quad$ For some $J \in \{I_1, \ldots, I_m\} \setminus \{I\}$ such that $\sum_{r \in J} f((r, v)) \notin \mathbb{Z}$, find $w \in J$ such that $(v, w) \in E_U$ and $f_U((v, w)) \notin \mathbb{Z}$
14: $\quad\quad\quad$ $D_1 \leftarrow D_1 \cup \{\sum_{r \in I} f((r, v))\}$ (corresponding to $(u, v)$)
15: $\quad\quad\quad$ $D_2 \leftarrow D_2 \cup \{\sum_{r \in J} f((r, v))\}$ (corresponding to $(v, w)$)
16: $\quad\quad$ **end if**
17: $\quad$ **else**
18: $\quad\quad$ Find $w \in V \setminus \{u\}$ such that $(v, w) \in E_U$ and $f_U((v, w)) \notin \mathbb{Z}$
19: $\quad$ **end if**
20: $\quad$ $C \leftarrow C \cup \{(v, w)\}$
21: $\quad$ $u \leftarrow v$
22: $\quad$ $v \leftarrow w$
23: **end while**
24: Set $e_1$ as the first edge in $C$ leaving $v$
25: Set $e_{-1}$ as the last edge in $C$ (entering $v$)
26: Remove edges preceding $e_1$ from $C$, and remove the corresponding elements from $D_1$ and $D_2$
27: **if** $v \in \mathcal{P}$ and $\exists I \in \{I_1, \ldots, I_m\}$ such that $e_1 \in \{v\} \times I$ and $e_{-1} \in I \times \{v\}$ **then**
28: $\quad$ Remove the elements corresponding to $e_1$ and $e_{-1}$ from $D_1$ and $D_2$
29: **end if**
30: **if** $e_1 \notin E$ **then**
31: $\quad$ Swap $D_1$ and $D_2$
32: **end if**
33: Replace each edge in $C$ from $E_U$ with the corresponding edge from $E$

---

- Line 9 is changed to:

$$\alpha \leftarrow \min\left(\min_{e \in A} f(e), \min_{e \in B} h(e) - f(e), \min_{t \in D_1} t - \lfloor t \rfloor, \min_{t \in D_2} \lceil t \rceil - t\right).$$

- Line 16 is changed to:

$$\beta \leftarrow \min\left(\min_{e \in A} h(e) - f(e), \min_{e \in B} f(e), \min_{t \in D_1} \lceil t \rceil - t, \min_{t \in D_2} t - \lfloor t \rfloor\right).$$

The primary modification we make to Algorithm 1 is replacing Line 6 with the subroutine in Algorithm 2. In each iteration, when we look for an undirected cycle of fractional edges in the graph, we now choose the cycle carefully rather than arbitrarily. We find a cycle by starting from an arbitrary fractional edge in the graph and walk along adjacent fractional edges (ignoring direction) until we repeat a previously-visited vertex. As we do this, whenever we take a fractional edge from a reviewer in subset $I$ into paper $p$, there are two cases.

- Case 1: If there exists a different fractional edge from paper $p$ to subset $I$ (Line 8 in Algorithm 2), we take this edge next. Note that if the total load from subset $I$ on paper $p$ is integral, such an edge must exist.

- Case 2: Otherwise (Line 12 in Algorithm 2), we must take a fractional edge from paper $p$ to some other subset $J$. In this case, the total load from subset $I$ on paper $p$ must not be integral. We choose the subset $J$ so that the total load from subset $J$ on paper $p$ is also not integral. Such a subset must exist since the total load on paper $p$ is always integral. We keep track of both the total load from $I$ and from $J$ on $p$, for every occurrence of this case along the cycle (Lines 14 and 15 in Algorithm 2).

In Case 1, no matter how much flow is pushed on the cycle, the total load from subset $I$ on paper $p$ will be preserved exactly. However, due to Case 2, we must modify the choice of how much flow to push on the cycle to ensure that the loads are preserved as desired. Specifically, we only push flow in a given direction on the cycle until the total load for either subset $I$ or $J$ on paper $p$ is integral, for any $I, J, p$ found in Case 2. The total loads from each subset on each paper found in Case 2 are saved in either set $D_1$ or set $D_2$ depending on the direction of the corresponding edges in the cycle, and each subset-paper pair with an edge corresponding to an element of $D_1$ or $D_2$ has only that one edge in the cycle. If the total (fractional) load from subset $I$ on paper $p$ is $t$, then only $\lceil t \rceil - t$ additional flow can be added to any edge from subset $I$ to paper $p$ before the load becomes integral; similarly, only $t - \lfloor t \rfloor$ flow can be removed from any edge before the load becomes integral. This leads to the stated changes to Lines 9 and 16 in Algorithm 1.

Therefore, on each iteration, we push flow until either the flow on some edge is integral (as in the original algorithm), or until the total load on some paper from some subset is integral. This implies that the algorithm still terminates in a finite number of iterations. In addition, by the end of the algorithm, the total load on each paper from each subset is preserved exactly if originally integral and rounded in either direction if originally fractional, as desired.

### F.2 Proof of Lemma 1 and Corollary 1

**Proof of Lemma 1:** We first prove part (i) of the lemma. Consider any subset $I$ and any paper $p$, and recall that in Section 5.1 we showed that the algorithm presented there has the property that the total load on each paper from each subset is preserved exactly if originally integral and rounded in either direction if originally fractional. If the total load from subset $I$ on paper $p$ is less than or equal to 1 originally (i.e., $\sum_{r \in I} F_{rp} \leq 1$), then this algorithm will only ever sample assignments with either 0 or 1 reviewers, so it never samples a integral assignment that assigns two reviewers from subset $I$ to paper $p$.

We now prove part (ii) of the lemma. Suppose that the total load from subset $I$ on paper $p$ is originally strictly greater than 1 (i.e., $\sum_{r \in I} F_{rp} > 1$). Let $X$ denote a random variable that represents the number of reviewers from subset $I$ on paper $p$, that is, $X = \sum_{r \in I} M_{rp}$. Hence, we have $\mathbb{E}[X] = \sum_{r \in I} F_{rp} > 1$. Suppose that we implement the marginal probabilities $F$ as a distribution over deterministic assignments that places zero mass on any deterministic assignment where $X \geq 2$. Since $X$ is integral in any deterministic assignment, all of the mass must be placed on deterministic assignments where $X \leq 1$. Since $\mathbb{E}[X] > 1$, this is impossible. Therefore, $F$ cannot be implemented without having some probability of placing two reviewers from subset $I$ on paper $p$, so the expected number of pairs of reviewers from subset $I$ assigned to paper $p$ must be non-zero for any sampling algorithm.

**Proof of Corollary 1:** We now show that the distribution sampled from by the algorithm realizing Lemma 1 minimizes the expected number of pairs of reviewers from each subset assigned to each paper. Consider any subset $I$ and paper $p$, and again let $X$ denote a random variable that represents the number of reviewers from subset $I$ on paper $p$. The expected number of pairs of reviewers from subset $I$ assigned to paper $p$ is $\mathbb{E}\left[\binom{X}{2}\right] = \frac{1}{2}\mathbb{E}[X^2] - \frac{1}{2}\mathbb{E}[X]$. Since $\mathbb{E}[X]$ is fixed for a given $F$, we must only show that our chosen decomposition minimizes $\mathbb{E}[X^2]$.

Let $f$ be the probability mass function of $X$ under the distribution of $X$ produced by our sampling algorithm, so that $f(i) = P[X = i]$ for $i \in \{0, \ldots, |I|\}$. Let $f'$ be the probability mass function of $X$ under any different distribution produced by some sampling algorithm, so that $\exists i \in \{0, \ldots, |I|\}$ such that $f'(i) \neq f(i)$. Since both $f$ and $f'$ are produced by sampling algorithms, they must respect the marginal assignment probabilities given by $F$.

First, assume that $\mathbb{E}[X] = \mu$ is integral. $\mathbb{E}[X] = \sum_{r \in I} F_{rp}$, so $\mu$ is equal to the total load from subset $I$ on paper $p$. From Section 5.1, our sampling algorithm preserves exactly the loads from any subset on any paper that are originally integral, meaning that it will always assign exactly $\mu$ reviewers from subset $I$ to paper $p$. In other words, our sampling algorithm always gives the distribution of $X$ where $f(\mu) = 1$ and $f(i) = 0$ for $i \neq \mu$. Since all distributions of $X$ have the same expectation, $\sum_{i=0}^{|I|} f'(i)i = \mu$; we also know that $f'(i) > 0$ for some $i \neq \mu$. For this distribution, we have that

$$\mathbb{E}_{f'}[X^2] = \sum_{i=0}^{|I|} f'(i)i^2 = \sum_{\Delta=-\mu}^{|I|-\mu} f'(\mu + \Delta)(\mu + \Delta)^2 = \mu^2 + \sum_{\Delta=-\mu}^{|I|-\mu} f'(\mu + \Delta)\Delta^2 > \mu^2 = \mathbb{E}_f[X^2].$$

Now, suppose that $\mathbb{E}[X] = \mu$ is not integral. From Section 5.1, our sampling algorithm rounds to a neighboring integer the loads from any subset on any paper that are originally not integral, meaning that it will always assign exactly $\lceil \mu \rceil$ or $\lfloor \mu \rfloor$ reviewers from subset $I$ to paper $p$. In other words, our sampling algorithm only places probability mass on outcomes $X = \lceil \mu \rceil$ or $X = \lfloor \mu \rfloor$, so $f(i) = 0$ for $i \notin \{\lceil \mu \rceil, \lfloor \mu \rfloor\}$. There is only one way to do this so that $\mathbb{E}[X] = \mu$; exactly $f(\lceil \mu \rceil) = \mu - \lfloor \mu \rfloor$ and $f(\lfloor \mu \rfloor) = \lceil \mu \rceil - \mu$. Then under this distribution, via some algebraic simplifications,

$$\begin{aligned}
\mathbb{E}_f[X^2] &= f(\lceil \mu \rceil)\lceil \mu \rceil^2 + f(\lfloor \mu \rfloor)\lfloor \mu \rfloor^2 \\
&= -\lceil \mu \rceil^2 + \lceil \mu \rceil - \mu + 2\lceil \mu \rceil \mu.
\end{aligned} \tag{19}$$

Under any other distribution of $X$ giving the probability mass function $f'$,

$$\mathbb{E}_{f'}[X^2] = \sum_{i=0}^{|I|} f'(i)i^2$$

$$= \sum_{\Delta=-\lceil \mu \rceil}^{|I|-\lceil \mu \rceil} \left( f'(\lceil \mu \rceil + \Delta)(\lceil \mu \rceil + \Delta)^2 \right) + 2\lceil \mu \rceil \sum_{\Delta=-\lceil \mu \rceil}^{|I|-\lceil \mu \rceil} \left( f'(\lceil \mu \rceil + \Delta)\Delta \right) + \sum_{\Delta=-\lceil \mu \rceil}^{|I|-\lceil \mu \rceil} \left( f'(\lceil \mu \rceil + \Delta)\Delta^2 \right)$$

$$= \lceil \mu \rceil^2 + 2\lceil \mu \rceil(\mu - \lceil \mu \rceil) + \sum_{\Delta=-\lceil \mu \rceil}^{|I|-\lceil \mu \rceil} \left( f'(\lceil \mu \rceil + \Delta)\Delta^2 \right). \tag{20}$$

We want to show that $\mathbb{E}_{f'}[X^2] > \mathbb{E}_f[X^2]$. From (19) and (20), it remains to show that

$$\sum_{i=0}^{|I|} f'(i)(i - \lceil \mu \rceil)^2 > \lceil \mu \rceil - \mu.$$

Note that because $f'(i) \neq f(i)$ for some $i$, there exists some $j \notin \{\lceil \mu \rceil, \lfloor \mu \rfloor\}$ such that $f'(j) > 0$. Further, $(i - \lceil \mu \rceil)^2 \geq (\lceil \mu \rceil - i)$ for all integers $i$ and $(i - \lceil \mu \rceil)^2 > (\lceil \mu \rceil - i)$ for all integers $i \notin \{\lceil \mu \rceil, \lfloor \mu \rfloor\}$. Therefore,

$$\sum_{i=0}^{|I|} f'(i)(i - \lceil \mu \rceil)^2 > \sum_{i=0}^{|I|} f'(i)(\lceil \mu \rceil - i) = \lceil \mu \rceil - \sum_{i=0}^{|I|} f'(i)i = \lceil \mu \rceil - \mu.$$

Therefore, $\mathbb{E}_{f'}[X^2] > \mathbb{E}_f[X^2]$ as desired, so $f$ is the probability mass function corresponding to the distribution of $X$ which minimizes $\mathbb{E}[X^2]$ (uniquely, since the inequality is strict). This concludes the proof that our algorithm minimizes $\mathbb{E}[X^2]$ and therefore minimizes the expected number of pairs from the same subset assigned to the same paper.

# G   Synthetic Simulations

We now present experimental results on synthetic simulations. All results are averaged over 10 trials with error bars plotted representing the standard error of the mean, although error bars are sometimes not visible since the variance is low. All experiments were run on a computer with 8 cores and 16 GB of RAM, running Ubuntu 18.04 and solving the LPs with Gurobi 9.0.2 [48].

(a) Pairwise-Constrained Problem

(b) Partition-Constrained Problem

(c) Runtime on Pairwise-Constrained Problem

Figure 3: Experimental results on synthetic simulations.

We consider two different simulations. First, we consider a simulated "community model" as used in past work [23]. In this model, $n = d = 360$ and $k = \ell = 3$; it is further parameterized by a group size $g$. For all $i \in \{0, g, 2g, \ldots, n\}$, reviewers $i$ through $i + g - 1$ have similarity 1 with papers $i$ through $i + g - 1$ and similarity 0 with all other papers. We consider four different group sizes $g$: 3, 6, 9, 12. We also consider a uniform random simulation, where each entry of the similarity matrix is independently and uniformly drawn from $[0, 1)$, fixing $n = d = 1000$ and $k = \ell = 3$.

In Figure 3a, we examine the performance of our algorithm for the Pairwise-Constrained Problem. For each simulation, we set all entries of $Q$ to a constant $q_0$ and observe the sum-similarity as we vary $q_0$ (on the x-axis). The objective value is reported here as a percentage of the optimal unconstrained solution's objective, as was done in Section 6. For the community models, the group size makes a large difference as to what an acceptable value of $q_0$ is. For example, with group size 6 and $q_0 = 0.5$, our algorithm will always assign all good reviewers to all papers; however, for any lower value of $q_0$ it can no longer do this and so the objective deteriorates rapidly. Note that since our algorithm is optimal, this deterioration is due to the problem being overconstrained for low values of $q_0$ and not due to an issue with the algorithm. For the uniform random simulation, our algorithm performs very well, since there are likely many reviewers with high similarity for each paper.

We also examine the performance of our algorithm for the Partition-Constrained Problem in Figure 3b. For each simulation, we fix $q_0 = 0.5$ and gradually loosen Constraint (6) in $\mathcal{LP}2$ by increasing the constant from 1 to 3 in increments of $0.2$, shown on the x-axis. We plot the sum-similarity objective of the resulting assignment, reported as a percentage of the optimal non-partition-constrained solution's objective (that is, the solution to the Pairwise-Constrained Problem with $q_0 = 0.5$). For the community model simulations, we assign all reviewers in each group to the same subset of the partition. Since all of the reviewers who can review each paper well are in the same subset, this presents a highly constrained problem (which our algorithm is solving optimally). As expected, our algorithm trades

Figure 4: Additional results for the effectiveness of bidding manipulation on ICLR, $\gamma = 4$.

off the number of same-subset reviewer pairs assigned to the same paper and the sum-similarity objective rather poorly (as would any other algorithm). Since $q_0 = 0.5$, there is no difference between the cases with group size 6 or greater. For the uniform random simulation, we assign random subsets of size 100. Since there are likely many reviewers with high similarity for each paper in different subsets, our algorithm again performs very well.

In Figure 3c, we show the runtime of our algorithm for the Pairwise-Constrained Problem on the various simulations, fixing $q_0 = 0.5$ and varying $n = d$ on the x-axis. The runtime of our algorithm is similar across the different simulations. Our algorithm solves the uniform random simulation case with $n = d = 5000$ in just over 10 minutes.

## H   Bidding Model and Additional Manipulation Experiment Results

In this section, we describe the methods used to add synthetic bids to the ICLR 2018 dataset for the experiment described in Section 6.2. We then present results from an additional experiment.

We first describe the bidding itself. Each reviewer $r$ chooses a bid $b_{rp} \in \{-1, 0, 1\}$ for each paper $p$, indicating "not interested," "neutral," or "interested" respectively. Based on the similarity function used in NeurIPS 2016 [27], we compute the final similarity between reviewer $r$ and paper $p$ as $S'_{rp} = \gamma^{b_{rp}} S_{rp}$, where $S_{rp}$ is the text similarity from the ICLR dataset and $\gamma$ is a fixed scale parameter. For the results presented in Section 6.2, $\gamma = 2$.

We now describe the model used to generate bids for the honest (non-malicious) reviewers. We divide the reviewers uniformly at random into three groups. The first group contains 20% of the reviewers, who all bid 0 on all papers. The second group contains 50% of the reviewers, who bid non-zero on a low number of papers. These reviewers consider each paper within the 10% of papers that have highest text similarity with them, and independently choose to bid non-zero on each one with probability 0.016. If a paper is selected to bid non-zero, the bid is chosen from $\{-1, 1\}$ with uniform probability. The third group contains 30% of the reviewers, who bid non-zero on a high number of papers. They follow the same bidding procedure as the second group, but bid with probability 0.24.

Finally, we present some additional results in Figure 4. In this experiment, we set scale parameter $\gamma = 4$. The results here are qualitatively similar to the results in Section 6.2 (which set $\gamma = 2$), except that the manipulation is even more powerful here. However, our randomized assignment algorithm still limits the success rate of the manipulation to the desired level of 0.5.