[Reviews · NeurIPS 2020]

Review 1

Summary and Contributions: This paper aims to improve the reviewer-paper matching algorithms that many computer science conferences use to assign reviewers to submitted papers. Most conferences currently employ a deterministic algorithm with a linear program at its core that maximizes the total match quality (sum of similarity scores) subject to load balancing constraints ensuring that no reviewer is assigned too many papers and every paper is assigned enough reviewers. A problem with a deterministic algorithm is that unethical reviewers can manipulate their similarity scores (either through bids or submitted features) in order to try to get assigned one particular paper in order to boost it or nuke it. Another problem with a deterministic algorithm is that it cannot be shared to the public without the public being able to reverse engineer the match and reveal the reviewers assigned to a paper. The authors show that both problems can be alleviated by going with a randomized algorithm. The authors define a natural generalization of the standard LP adding constraints that every reviewer-paper assignment has probability less than some constant, like 50%. The authors include a clever way to convert a randomized assignment into an actual assignment that satisfies the load-balance constraints. The authors show that restricting the probability of a three-way match (two reviewers assigned to one paper) is NP-hard. The authors show that a special case where the probability is zero of a three-way match where the two reviewers are from the same institution (or the same set among any partition of the reviewers) can be solved in polytime. The authors run a number of experiments on a real conference data set (ICLR 2017) and simulated data. The results are promising: even when reviewer-paper matches happen with at most 50% probability, the overall match quality is 90% as good as the deterministic version.

Strengths: The basic idea of the paper is a very good one. Conferences SHOULD use random assignments. There seems to be very little downside. The paper is very well written. To me, the largest contributions are conceptual and empirical. The theoretical results are not as important but still help frame the problem space. The algorithm to convert a random assignment to a deterministic assignment is clever. The empirical results are compelling and promising although from only one real conference.

Weaknesses: The basic idea is simple. I actually view that as a strength. The NP-hardness proof is not all that useful. If every individual has at most a 50% chance to be assigned a paper, then every pair has at most a 25% chance, even without the extra constraint on 3-way matches. In reality other constraints besides load-balance constraint are needed, including conflict-of-interest constraints. In reality, the conference chairs do a lot of manual tuning, trading, changing, adjusting scores, etc. The authors don't directly model reviewer incentives. What level of randomness is good? With a model of reviewer utility, we may be able to suggest an answer. The authors assume similarity scores are given. In reality, manipulation probably happens more in the bidding than in the TPMS data or feature data. A system may want to focus on guarding the influence of bids. The authors assume the similarity scores are "ground truth". A better measure would be to actually survey authors to assess the quality of the match.

Correctness: Yes.

Clarity: The paper is very well written. Some comments, questions, and suggestions: Line 376: "use of these algorithms will likely result in slightly poorer paper assignments in terms of reviewer fit" >> I'm not so sure. The reviewer-paper similarity scores are approximations. Reviewers may not even notice a 90% reduction in similarity score. It would be interesting to test this by asking reviewers to rate the match quality. I suspect there is significant noise between the similarity scores and the relevance scores of reviewers. Even asking a reviewer to rate the match quality on two different days could result in noise/variance of 10% from one day to the next. So reducing the similarity scores may not be all that bad. Where are conflict-of-interest constraints in the LP? These show disallow certain matches with 100% probability. Do the authors allow this? Does this change the computational complexity of the problem? Line 65: "Although these challenges may seem disparate" >> challenges (1) and (2) seem the same and perhaps can be combined, though I agree (3) seems different. Line 174: I assume the "simpler version of the sampling algorithm" ensures that the load-balance constraints are met every time, not just in expectation? Line 365: "known network of reviewers" >> presumably if the network were known, conference organizers would remove them from the program committee. There are capitalization errors in the References section

Relation to Prior Work: I don't know this literature but the coverage of related work seems low. There is work in recommender systems about detecting and evaluating false reviews, for example: https://dl.acm.org/doi/abs/10.1145/988672.988726 . The same questions arise and have been studied on platforms like Yelp, Amazon, web search, etc., where research try to find methods to identify and discount dishonest or collusive behavior.

Reproducibility: Yes

Additional Feedback: Update: I read the authors' response which was well done and helpful.


Review 2

Summary and Contributions: The authors propose a randomized framework for the assignment of papers to reviewers. The goal is to overcome or mitigate some limitations detected in current revision processes (untruthful favorable reviews, torpedo reviewing, and reviewer de-anonymization in releasing assignment data).

Strengths: The paper is well-written, and the topic tackled is relevant for the peer-reviewing scientific process. Up to my knowledge, the proposal is novel and the contribution is reasonable.

Weaknesses: In my opinion, possibly the main limitation of this work is the absence of competitor methods, i.e. the authors do not compare with any other prior method in the literature. Another possible limitation is that the proposed algorithm is a bit hidden between all this verbosity. Maybe the explicit introduction of an algorithm would help in this regard.

Correctness: Yes, I think so.

Clarity: Yes, it's well written.

Relation to Prior Work: Yes, but it's also true that the related literature section is a bit short (specially taking into account the existing amount of work in the reviewer assignment problem).

Reproducibility: Yes

Additional Feedback: I have a somewhat general question for the authors. My question is how can we evaluate that their assignment framework actually reduces untruthful favorable reviews and torpedo reviewing. How could we empirically verify that their proposal quantitatively reduces both challenges? Why the authors did not compare with any other prior approach in the literature? ** AFTER READING RESPONSE ** After reading all reviewers' comments and the authors' feedback, I still consider it would be reasonable to accept this paper but, in any case, I don't strongly believe it should be accepted. The reviewers' comments about the formalization of the objectives (and the other weaknesses identified) are sensible and, taking into account all the information, my final mark would be between 6 and 7.


Review 3

Summary and Contributions: This paper formulates and gives an efficient solution to the problem of probabilistic peer review assignment to prevent strategic manipulation aimed at raising or lowering the rating of a particular paper in peer review, as well as de-anonymizing reviewers' identities from published review data. It also proves the difficulty of this problem under general constraints and proposes an extension of the proposed algorithm to the case where no one belonging to the same group can be assigned to the same paper. Experimental results using real data shows the promise of the proposed method.

Strengths: Once the problem setup is accepted, the proposed approach is properly designed and well analyzed.

Weaknesses: While it makes intuitive sense that probabilistic assignments can weaken strategic manipulation and de-anonymization, it is not clear to what extent the level of randomization specifically affects them.

Correctness: I was unable to confirm the correctness of the details of the proof, but the claims of the methods and theorems sound convincing. In the experiments, it is encouraging that increasing the randomization level does not result in a significant loss of similarity. On the other hand, it is still not clear to what extent this prevents strategic manipulation and de-anonymization.

Clarity: This paper is relatively well-written. Although there is too much contents to include in the allowed pages, the details of the algorithm, the proofs of theorems, and other important parts are pushed into the appendix, which blurs the most technically important part of this paper.

Relation to Prior Work: Yes.

Reproducibility: Yes

Additional Feedback: As mentioned above, the solution and analysis are fine, but the relationship between the original goal and the problem formulation as well as the benefit of the proposed approach on the original goal should be discussed.


Review 4

Summary and Contributions: In the assignment problem considered, each reviewer is assigned at most k papers and each paper is assigned exactly l (ell) reviewers. A matrix S represents similarities between reviewers and papers, and the goal is to maximize total summed similarity. The paper proposes a randomized assignment and introduces a constraint matrix Q with the maximum probability of assigning each reviewer to each paper. It solves the problem by solving the LP to get a fractional assignment, then devising an algorithm (an improvement on prior work[4]) for sampling an assignment in a way consistent with that solution. It then considers pairwise constraints, i.e. Q specifies that maximum probability that a given pair of reviewers are simultaneously assigned to a given paper. It shows this to be NP-hard, but can efficiently solve the 0-1 case, e.g. disallow multiple reviewers from the same institution. Finally, the paper runs experiments on the ICLR reviewing dataset, and a PrefLib dataset, to explore the tradeoff between stricter constraints and assignment quality (measured by the LP objective). These suggest that pretty high levels of randomization can be achieved without much loss in quality.

Strengths: It addresses a natural problem. It seems to be novel. The idea of randomizing review assignments is nice, if not surprising. The solution is promising for practical implementations.

Weaknesses: None of the following weaknesses is severe, but they add up somewhat. I would say the problem formulation and solution are only moderate contributions, because at a high level, one can summarize some fraction of the contributions as "formulate as an LP and sample from it using [4]". Not all the contributions by any means, of course. The submission does not really contain machine learning, which is the topic of the conference. I would have expected this topic at a general AI conference instead. The submission could do more to formalize the sense in which its objectives are accomplished. See comments to authors. The flow-based algorithms are discussed only at a very high level - they seem interesting, but too bad they were not discussed in detail.

Correctness: From the high-level descriptions in the main paper, I agree the claims are likely to be correct. I liked the empirical methodology, except that the number of trials per data point was a bit small (10, I believe). But I appreciated the tradeoffs explored in the experiments.

Clarity: Yes, I found it well-written.

Relation to Prior Work: Yes, I think so. In this case it is difficult because the underlying mathematical problem is a pretty general LP-based random sampling question so it's hard to know if it's been studied at some point.

Reproducibility: Yes

Additional Feedback: Regarding rigorously achieving the objectives: I feel the paper hand-waves a bit how randomization solves their objective problems. The objectives can be broken down into anonymity, and strategic manipulation. Anonymity: While I agree intuitively that adding randomness can help with anonymity, it would be nice to formalize. One approach could be differential privacy. Strategic manipulation: My reply to lines 124-127: But moving to randomized assignments doesnt' at all eliminate incentives or effectiveness of strategic manipulation. It might limit it a bit -- e.g. manipulation can only raise the probability of assignment from 0.3 to 0.8, or something. This is better than a deterministic setting where it can be raised from 0 to 1. But people still might want to manipulate to raise their chances. Maybe just a bit more discussion or formalization, tying the solution back to the objectives, would help readers feel these have been accomplished. One aspect the paper doesn't address is manipulation of the similarities, e.g. bidding higher or biasing input data (like published-paper database) in order to seem more similar to a paper. It is very clean to abstract this into an S matrix, but it would be interesting to discuss. -------- After author response: Thanks for the response. I agreed with the points at a high level, i.e. the idea that these points could be formalized. At the same time, I feel that until they're formalized in the paper or investigated more, the weaknesses remain. I like the author response as suggesting directions to formalize.

[Author Response · NeurIPS 2020]

We thank all of the reviewers for their comments and suggestions.

**Reviewers 2, 3, and 5** asked how well our proposed algorithms address the identified challenges. For the challenges
(i) and (ii) involving reviewer manipulation of the assignment, our problem formulations *consider the worst-case*
*manipulation*; that is, we want to effectively address the challenges even if a malicious reviewer could somehow always
get themselves assigned to their desired paper under a standard (deterministic) assignment algorithm. In the worst
case, the maximum probability of that reviewer-paper assignment given as input to our algorithm *exactly specifies how*
*well we address the manipulation*, since it is the probability that the manipulation works (reduced from 100% in the
deterministic case). This approach allows us to *avoid making assumptions* on what reviewers are willing to do that may
not hold in practice; for example, reviewers have been known to fully falsify reviewer profiles to get a desired paper
assignment. For the de-anonymization challenge (iii), the maximum probability of a reviewer-paper assignment given
as input to our algorithm is exactly the highest certainty that an author can have that a specific reviewer reviewed their
paper (assuming the review contains no identifying information), *specifying how well we address this challenge*.

**Reviewers 1 and 5** raised similar points regarding alternate approaches to the problem that more explicitly model
reviewers' manipulation of similarity scores (e.g., reducing the influence of bids in the similarity computation).
Modeling reviewers' manipulation of similarities more explicitly *would require making some assumptions* on reviewer
behavior/incentives (which may be unlikely to hold in practice) as well as on the similarity computation used (which
varies across conferences). Our approach to the problem **does not make any such assumptions** and instead provides
worst-case guarantees as described above.

**Reviewer 1:**

• *Regarding surveying reviewers to determine the ground truth assignment quality:* We agree with you that
similarities are a noisy approximation. However, similarities are the standard measure of assignment quality
used in past work as well as in practice (including at NeurIPS 2020!), and the question of better evaluating
assignment quality even in the absence of malicious behavior is open. Additionally, since we compare our
proposed assignment to the standard deterministic assignment and the similarities are used to approximate
the assignment quality for both assignments, any noise in the similarities should not impact our evaluation
significantly.

• *Regarding conflict-of-interest constraints:* As we mention in Line 115 of the paper, conflicts-of-interest can
be incorporated into the similarity matrix by assigning a similarity of $-\infty$ to any reviewer-paper pair with a
conflict of interest, so conflict-of-interest constraints do not change the problem and all results go through.

• *Regarding the "simpler version of the sampling algorithm" (Line 174):* The algorithm does ensure that the
load balance constraints are met every time.

• *Regarding the additional line of related work:* We appreciate your suggestion, and will expand our coverage
of related work in any revisions of the paper.

**Reviewer 2:**

• *Regarding an empirical evaluation of our algorithm's effectiveness at addressing reviewer manipulation:* In
real-world conferences, reviewers do not reveal the ground truth of whether they engaged in manipulation, so it
is not possible to empirically evaluate the extent of reduction in manipulation. This motivates our "worst-case"
view on manipulation described above where we aim to optimally mitigate manipulations no matter what the
adversary does.

• *Regarding comparing our algorithm's performance against prior work:* To clarify, we theoretically show that
our algorithms **optimally** solve the problems they are designed for. Furthermore, there is no prior literature
in peer review addressing the challenges we identify, and the literature in other fields such as recommender
systems requires additional data or assumptions that are unavailable or inapplicable here. Hence, our empirical
evaluation compares our algorithm against the current method used in practice (including in NeurIPS 2020), a
deterministic assignment algorithm.

**Reviewer 5:**

• *Regarding reviewer incentives for manipulation:* It is impossible to entirely remove incentives for reviewers
to behave maliciously unless we make strong additional assumptions or have access to some information
exogeneous to the conference (e.g., external communication between reviewer and author). Since we cannot
entirely prevent manipulation in our setting, we aim to mitigate it as much as possible.

• *Regarding the use of the algorithm from [4]:* We would like to clarify that we optimize and simplify the
algorithm from [4] significantly for our setting.

[Meta-Review · NeurIPS 2020]

The reviewers were initially fairly positive about the contribution of this paper. Some of the highlights are that the proposed method seems simple and practical (and, e.g., could be used right away for conference matching). During the discussion, two issues were raised: 1) The lack of formalization of the objective (this is detailed in Reviewer #5's review and post-response comments). The authors provided a very good response in their rebuttal and while some of these elements could be added to the paper, there was a consensus that a deeper investigation is warranted. 2) While this paper is interesting, it is somewhat less relevant to the core interests of NeurIPS (although it falls within the topics outlined in the CFP) and may receive more interest from the participants at other venues (e.g., IJCAI and AAAI). Also, I wanted to thank you for the AC-note you sent with your response. I agree that while both real-world evaluations and surveying the participants of a conference could add a lot of value to this work, both constitute major endeavors that are better left for future projects. Our discussion did not center around these aspects and so I believe that the reviewers agreed with your response.